# Generalization in Online Reinforcement Learning for Mobile Agents

## Abstract

Graphical user interface (GUI)-based mobile agents automate digital tasks on mobile devices by interpreting natural-language instructions and interacting with the screen. While recent methods apply reinforcement learning (RL) to train vision-language-model(VLM) agents in interactive environments with a primary focus on performance, generalization remains underexplored due to the lack of standardized benchmarks and open-source RL systems. In this work, we formalize the problem as a Contextual Markov Decision Process (CMDP) and introduce **AndroidWorld-Generalization**, a benchmark with three increasingly challenging regimes for evaluating zero-shot generalization to unseen task instances, templates, and applications. We further propose an RL training system that integrates Group Relative Policy Optimization (GRPO) with a scalable rollout collection system, consisting of containerized infrastructure, asynchronous execution, and error recovery to support reliable and efficient training. Experiments on AndroidWorld-Generalization show that RL enables a 7B-parameter VLM agent to surpass supervised fine-tuning baselines, yielding a 26.1% improvement on unseen instances but only limited gains on unseen templates (15.7%) and apps (8.3%), underscoring the challenges of generalization. As a preliminary step, we demonstrate that few-shot adaptation at test-time improves performance on unseen apps, motivating future research in this direction. To support reproducibility and fair comparison, we open-source the full RL training system, including the environment, task suite, models, prompt configurations, and the underlying infrastructure.

## 1 Introduction

Mobile agents are autonomous digital systems that control mobile devices via natural language to automate tasks (Wu et al., 2024; Liu et al., 2025a). Unlike API-based agents limited to predefined function calls (Song et al., 2024; Zhang et al., 2025), graphical user interface (GUI)-based agents interact directly with the screen through actions such as clicking and typing, enabling broader applicability across diverse apps and devices (Gou et al., 2025; Qin et al., 2025). Developing GUI-based mobile agents is challenging: they must interpret instructions, handle diverse screenshots, and plan coherent multi-step actions across apps in dynamic environments.

Inspired by recent advances in the reasoning capabilities of large vision–language models (VLMs) (Jaech et al., 2024; Comanici et al., 2025), several studies leverage prompting techniques on proprietary VLMs to construct predefined decision-making pipelines (Li et al., 2024b; Agashe et al., 2025; Li et al., 2025). An alternative direction is post-training an open-source VLM on offline static trajectory datasets tailored to mobile scenarios, using either human-annotated or synthetically generated trajectories via supervised fine-tuning (Qin et al., 2025; Sun et al., 2025). However, static datasets cannot capture the full interactive dynamics of mobile environments, leading agents to suffer from error accumulation and poor generalization to unseen environment changes (e.g., UI variations and dynamics) (Bai et al., 2024). To address these limitations, recent work explores online reinforcement learning (RL) in interactive environments. In this setting, mobile agents are trained with VLM-based policies that generate multi-step trajectories, while optimizing their behavior from online reward feedback provided by the environment (Bai et al., 2024; Papoudakis et al., 2025; Gu et al., 2025; Shi et al., 2025).

Figure 1: Sample task instructions with corresponding screenshots from the train and test sets of the three unseen regimes in the AndroidWorld-Generalization benchmark. Red highlights the unseen scenarios: Instance, Template, and Application.

Despite recent progress, prior online RL methods for mobile agents still face a fundamental limitation: they primarily focus on algorithmic improvements to boost performance on standard benchmarks, while generalization remains largely underexplored. In practice, however, mobile agents must operate robustly in dynamic, open-ended environments and handle unseen scenarios such as novel tasks, unfamiliar UI layouts, or entirely new applications. First, this limitation stems largely from the fact that existing benchmarks are designed solely for evaluation and do not provide a designated training set. Consequently, prior works either train and test on the same evaluation tasks (Papoudakis et al., 2025), which ignores assessment of generalization, or construct synthetic training tasks without verifying the absence of train–test leakage (Shi et al., 2025; Yang et al., 2025). This lack of a principled train–test split makes it difficult to systematically study generalization.

Second, the field lacks an open-source RL training system for realistic mobile environments, which limits reproducibility and fair comparison. Existing works are either closed-source or release only model weights, even though agent performance also depends on prompt templates, agent logic, and training recipes. In addition, building a reliable and efficient RL system for realistic mobile environments poses significant engineering challenges, as the environments are computationally expensive, delay-prone, and crash-sensitive. These challenges have created a substantial barrier between conceptual advances in LLM-based RL and their practical realization in mobile environments. Without such a reliable training system, progress on algorithmic innovation is fundamentally constrained.

In this work, we study generalization in online RL for mobile agents. We first formalize the mobile environment as a Contextual Markov Decision Process (CMDP) (Hallak et al., 2015) and evaluate generalization through zero-shot policy transfer. Each context defines a distinct MDP (e.g., a task instance, a task template, or an application), allowing training on one set of contexts and evaluation on previously unseen ones. We adopt AndroidWorld (Rawles et al., 2025) as both the training environment and testbed, since it provides rule-based scripts for reliable rewards and an automatic task-parameterization mechanism that generates thousands of diverse task instances for constructing held-out contexts. Building on this, we introduce *AndroidWorld-Generalization*, which defines three progressively challenging regimes: Unseen Instance, Unseen Template, and Unseen Application. Second, we develop the first fully open-source RL training system for mobile agents, integrating Group Relative Policy Optimization (GRPO) (Guo et al., 2025) with an Android emulator environment. To support large-scale and reliable RL, we design a scalable rollout collection system consisting of a containerized infrastructure that provides resource isolation via Docker, asynchronous rollout execution to eliminate synchronization bottlenecks, and robust error-recovery mechanisms for handling emulator failures. Together, these components enable efficient, stable, and scalable RL training in realistic mobile environments.

Extensive evaluation on AndroidWorld-Generalization shows that online RL enables a 7B-parameter VLM mobile agent to surpass supervised fine-tuning baselines by 26.1% and even outperform proprietary model–based pipelines such as GPT-4o and Claude Computer Use. However, the challenges of generalization remain, with limited gains in unseen templates (15.7%) and unseen apps (8.3%).

Finally, we demonstrate that using few-shot adaptation at test-time can improve performance by 10.4% on the most challenging Unseen App setting. In summary, our contributions are as follows:

- We present the first study of generalization in RL for mobile agents by formalizing the problem as a Contextual MDP and introducing **AndroidWorld-Generalization**, a benchmark with three progressively challenging regimes for evaluating zero-shot policy transfer.
- We develop the first fully open-source RL training system for mobile agents, integrating GRPO with a scalable rollout collection system that ensures reproducibility and offers infrastructure for future work.
- We conduct the first empirical study of RL generalization in mobile agents, demonstrating strong performance on unseen instances but limited transfer to templates and apps, and identify few-shot adaptation at test-time as a promising direction.

## 2 RELATED WORKS

**GUI Mobile Agents.** Prior works can be broadly classified into three categories. Prompting-based approaches construct predefined decision-making pipelines—covering perception, memory, and planning—by orchestrating multiple proprietary VLMs (Wen et al., 2024; Li et al., 2024b; Wang et al., 2024a; 2025c; Agashe et al., 2025; Li et al., 2025), but incur high cost and limited adaptability. Offline methods encode domain-specific capabilities by post-training a single VLM on large-scale human-annotated (Qin et al., 2025) or synthetically generated trajectories (Wu et al., 2025b; Sun et al., 2025; Gandhi & Neubig, 2025; Bai et al., 2025a), typically relying on static benchmarks (Li et al., 2020; Sun et al., 2022; Hsiao et al., 2022; Rawles et al., 2023; Li et al., 2024a; Wang et al., 2024c; Chai et al., 2025a). However, offline datasets restrict evaluation to per-step accuracy and cannot capture trajectory-level, long-horizon success (Pan et al., 2024). Online RL methods allow agents to interact with dynamic environments and optimize VLM-based policies from reward feedback (Liu et al., 2024; Bai et al., 2024; Wang et al., 2025b; Papoudakis et al., 2025; Gu et al., 2025; Shi et al., 2025; Yang et al., 2025). These approaches are typically evaluated on interactive benchmarks (Wang et al., 2024a; Xing et al., 2024; Chai et al., 2025b; Chen et al., 2025; Rawles et al., 2025; Xu et al., 2025), which provide Android emulator platforms (Toyama et al., 2021) and assess trajectory-level success via rule-based scripts or LLM-as-a-judge (Gu et al., 2024; Lù et al., 2025). However, these benchmarks lack standardized training environments and unseen contexts, limiting systematic study of RL generalization in mobile agents.

**Reinforcement Learning for LLM Agents.** Reinforcement learning (RL) has proven effective for fine-tuning LLMs on reasoning tasks (Guo et al., 2025; Team et al., 2025; Ke et al., 2025) and has recently been extended to multi-turn agentic decision-making. Early works align LLMs with textual embodied environments through online RL (Carta et al., 2023; Tan et al., 2024; Zhou et al., 2024b; Wang et al., 2025d), while Zhai et al. (2024) apply LoRA with PPO (Schulman et al., 2017) to fine-tune 7B VLMs, surpassing GPT-4V and Gemini. More recent studies bring RL into realistic GUI-based environments, particularly in web and computer-use domains (Qi et al., 2025; Wei et al., 2025; Wu et al., 2025a; Vattikonda et al., 2025; Lu et al., 2025; Feng et al., 2025). By contrast, work on mobile agents remains limited: offline methods rely on static curated datasets that fail to capture full environment dynamics (Luo et al., 2025; Lu et al., 2025; Liu et al., 2025c; Bai et al., 2025a), while online approaches explore interaction-based training. For example, Bai et al. (2024) propose an offline-to-online framework, Shi et al. (2025) and Gu et al. (2025) extend GRPO to trajectory-level optimization with customized rewards, and Yang et al. (2025) automate task generation and reward estimation. However, those approaches primarily target performance improvements while overlooking generalization to unseen scenarios. On the system side, distributed and scalable RL systems (Wang et al., 2025b; Lai et al., 2025) remain largely closed source, limiting reproducibility.

**Benchmarks for Generalization in Reinforcement Learning.** Generalization in reinforcement learning is the ability of agents to transfer robustly to unseen environments, commonly formalized as a Contextual Markov Decision Process (CMDP) (Hallak et al., 2015), where training and testing occur on disjoint context sets (Kirk et al., 2023). Benchmarks for this problem typically adopt either *procedural generation*, where distinct context are sampled from random seeds (Cobbe et al., 2019; 2020; Küttler et al., 2020; Team et al., 2021; Chevalier-Boisvert et al., 2023), or *controllable variation*, where environment parameters such as states, dynamics, or reward functions are

explicitly configured to enable systematic evaluation (Packer et al., 2018; Ahmed et al., 2020; Zhu et al., 2020; Hansen & Wang, 2021; Benjamins et al., 2021). Recent benchmarks targeting real-world tasks have been proposed in domains such as web navigation (Yao et al., 2022; Zhou et al., 2024a; Koh et al., 2024), computer use (Bonatti et al., 2025; Xie et al., 2024), and enterprise work-flows (Drouin et al., 2024), but they primarily provide evaluation sets without standardized train–test splits. While Liu et al. (2025b) augment these benchmarks with additional test sets, they still fail to capture environmental variability. A concurrent work examines generalization through factors such as icon placement, size, wallpapers, languages, and device types in mobile environments, but does not incorporate reinforcement learning (Lee et al., 2025).

## 3 BENCHMARKING GENERALIZATION IN MOBILE ENVIRONMENTS

Existing benchmarks focus on evaluation without standard training data, while prior methods often rely on human-collected or synthetic data that is rarely released. This lack of transparency raises con-cerns about potential train–test leakage and limits the study of generalization. To address this gap, we formalize mobile interactions as a Contextual Markov Decision Process (CMDP) and introduce a new benchmark, AndroidWorld-Generalization, designed to systematically evaluate generalization.

**Preliminaries.** A common formalism for mobile interactions is the Markov Decision Process (MDP) $\mathcal{M} = \langle \mathcal{S}, \mathcal{A}, \mathcal{T}, \mathcal{R} \rangle$. The state space $\mathcal{S}$ consists of GUI screenshots combined with in-teraction history. The action space $\mathcal{A}$ comprises mobile interactions such as clicking coordinates, swiping or typing variable-length text, etc. The transition function $\mathcal{T}$ is determined by the mobile operating system, and the reward function $\mathcal{R}$ provides a binary signal at the terminal state, indicat-ing task success or failure. At each timestep $t$, given a natural-language task instruction $q$, the agent takes the current state $s_t$ as input and selects an action $a_t \in \mathcal{A}$. The state is defined as $s_t = (o_t, h_t)$, where $o_t$ denotes the current observation (screenshot) and $h_t = \{o_0, a_0, o_1, a_1, \ldots, o_{t-1}, a_{t-1}\}$ is the interaction history recording all past observations and actions. The policy is parameterized by a VLM as $a_t \sim \pi_\theta(\cdot \mid s_t, q)$. In practice, rewards in mobile environments are *sparse and terminal-only*: the agent always receives $r_t = 0$ except a binary signal $r_T \in \{0, 1\}$ at the terminal step.

### 3.1 GENERALIZATION IN CONTEXTUAL MDP

However, the standard MDP assumes a single stationary environment and thus cannot capture vari-ability across tasks. To model diverse tasks and enable the study of generalization, we extend MDP into a *Contextual Markov Decision Process (CMDP)* (Hallak et al., 2015). A CMDP factors the state space as $\mathcal{S} = \mathcal{S}' \times \mathcal{C}$, where $\mathcal{S}'$ is the underlying state space and $\mathcal{C}$ is a context space. A context $c \in \mathcal{C}$ captures higher-level variations, such as different task instructions within a template, different templates within an application, or entirely different applications. Before each interaction sequence, a context $c \sim P_\mathcal{C}$ is sampled from a distribution over $\mathcal{C}$ and remains fixed until the task is completed. The context influences both states and transitions. For example, if $c$ corresponds to the Google Calendar, then states include UI screenshots, while transitions correspond to operations such as creating a new event or setting reminders, rather than those from other applications.

To evaluate generalization, we adopt the *zero-shot policy transfer (ZSPT)* (Kirk et al., 2023). Specifi-cally, we partition the context space into two disjoint subsets, $\mathcal{C}_{\text{train}}$ and $\mathcal{C}_{\text{test}}$, such that $\mathcal{C}_{\text{train}} \cap \mathcal{C}_{\text{test}} = \emptyset$, sampled from the same underlying distribution $P_\mathcal{C}$. The agent is trained only on $\mathcal{C}_{\text{train}}$, but its perfor-mance is evaluated on $\mathcal{C}_{\text{test}}$. With terminal-only rewards, the objective becomes

$$\max_{\pi_\theta} \mathbb{E}_{c \in \mathcal{C}_{\text{test}}}[r_T \mid c],$$

without any additional training or fine-tuning on $\mathcal{C}_{\text{test}}$.

### 3.2 ANDROIDWORLD-GENERALIZATION

**Why AndroidWorld?** To instantiate the CMDP formulation in realistic mobile environments, we extend AndroidWorld into a new benchmark, AndroidWorld-Generalization, designed to enable fair and reproducible evaluation of zero-shot generalization, shown in Figure 1. Although AndroidWorld was originally introduced solely for evaluation rather than RL training, it has two properties that

Table 1: **Statistics of the AndroidWorld-Generalization benchmark**. "I", "T", and "A" denote Instances, Templates, and Applications; "Diff." denotes average difficulty. **Bold numbers** indicate the number of train/test examples in each regime, while gray numbers indicate the templates and applications from which those examples are generated.

| Regime | Overlap | | | Train | | Test | |
|---|---|---|---|---|---|---|---|
| | Instance | Template | App | I/T/A | Diff. | I/T/A | Diff. |
| Unseen Instance | ✗ | ✓ | ✓ | 1149 / 78 / 17 | 1.68 | 234 / 78 / 17 | 1.68 |
| Unseen Template | ✗ | ✗ | ✓ | 836 / 57 / 14 | 1.70 | 54 / 18 / 14 | 1.72 |
| Unseen App | ✗ | ✗ | ✗ | 905 / 62 / 12 | 1.68 | 48 / 16 / 5 | 1.69 |

make it well suited for extension into an RL benchmark. First, it provides rule-based scripts that ensure reliable reward functions rather than relying on LLM-as-a-judge. Detailed comparisons are given in Appendix G. Second, its automatic task parameterization mechanism enables the generation of thousands of diverse task instances from 116 templates, which span three difficulty levels across 20 applications, thereby facilitating the systematic construction of held-out contexts.

This task parameterization mechanism induces a natural hierarchy for task instance generation: each application contains multiple task templates, and each template can produce many distinct instances by sampling different random seeds. For example, in the Markor note-taking application, the template "*Create a new note named {file_name} with the following text: {text }*" contains two parameters, and varying these parameters under different seeds produces different task instances.

**Train-evaluation task generation.** Building on this parameterization, we define *task instances*, *task templates*, and *applications* as distinct notions of *context* in AndroidWorld-Generalization, and introduce three challenging regimes: In **Unseen Instance**, we begin with all 116 AndroidWorld templates but discard 38 that cannot generate distinct task instances using random seeds, leaving 78 usable templates. We then construct the splits by generating evaluation instances using 3 fixed seeds and training instances using 16 non-overlapping seeds. This yields 1149 unique training instances and 234 test instances, while keeping the same templates and applications shared across splits. In **Unseen Template**, we first filter out applications that contain only a single template, then partition the remaining templates within each app under a 3:1 train–test ratio. This yields 57 training templates and 18 held-out templates drawn from 14 shared applications. After defining the template split, we generate task instances using the same procedure as in the Unseen Instance regime, assigning non-overlapping sets of random seeds to obtain 836 distinct training instances and 54 testing instances. In **Unseen App**, we construct a fully disjoint application split, using 12 applications for training and 5 distinct applications for testing, such that applications, templates, and task instances are all non-overlapped. For example, the agent is trained on tasks from Calendar and evaluated on tasks from Camera. In all regimes, training and evaluation are conducted on disjoint context sets.

To ensure fairness, we balance task difficulty across the training and testing sets and manually verify that no task instances overlap. Full details of task-instance generation, train–test split construction, difficulty computation, and the differences between AndroidWorld-Generalization and the original AndroidWorld are provided in Appendix C, with benchmark statistics summarized in Table 1. We report average test-set success rates with standard deviation across three evaluation seeds per task template.

## 4 TRAINING SYSTEM FOR MOBILE AGENTS

Building on the established generalization benchmark, we now describe the training of mobile agents in this setting. In this section, we present the design of mobile agents trained with Group Relative Policy Optimization (GRPO) as the online reinforcement learning algorithm. We further analyze the limitations of the native AndroidWorld implementation and develop a scalable rollout collection system to enable reliable and efficient training.

### 4.1 ONLINE LEARNING WITH GRPO

Mobile agents must perform multi-turn decision-making through interaction with mobile environments. To accommodate the resource constraints of mobile devices, we adopt Qwen2-VL-7B ar-

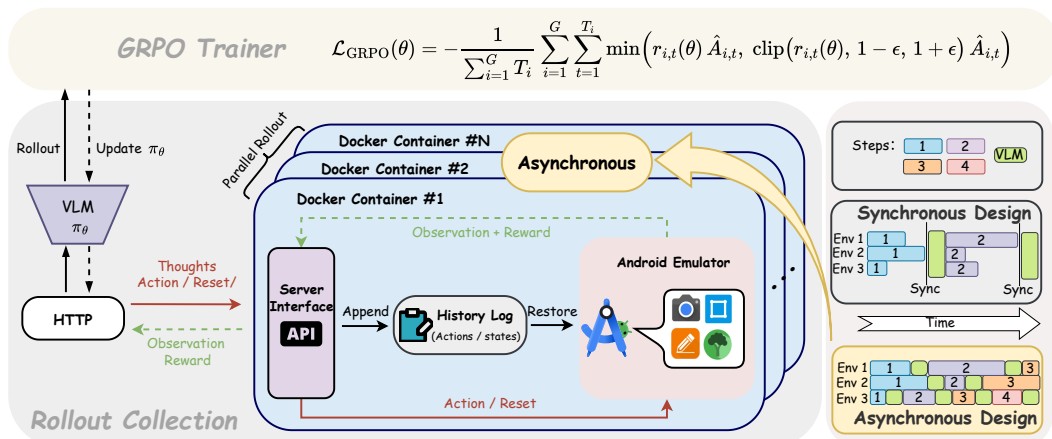

Figure 2: **RL training system for mobile agent.** We integrate GRPO with a scalable rollout collection system that parallelizes multiple environments. Docker containerization provides resource isolation and decouples trainer and environments through HTTP communication for reliability. Asynchronous rollouts eliminate synchronization bottlenecks, enabling more agent steps per unit time, while error recovery mechanisms resume rollouts from failures without restarting. Together, these three techniques facilitate reliable and efficient large-scale training.

chitecture as the policy model (Wang et al., 2024b). We initialize it with UI-TARS (Qin et al., 2025) weights, supervised fine-tuned on large-scale annotated GUI trajectories. This initialization can provide domain-specific priors that serve as an effective warm start for reinforcement learning (Zhai et al., 2024). At each timestep of a trajectory, the agent conditions on the current screenshot and the full interaction history to capture long-term dependencies. Inspired by Zhai et al. (2024), we incorporate chain-of-thought prompting (Wei et al., 2022) to enhance reasoning, structuring outputs into *thoughts* and *actions*, while the interaction history records all preceding screenshots, thoughts, and actions. The detailed prompt template is provided in Appendix H.

We adopt GRPO from DeepSeek-R1 (Guo et al., 2025) as our online reinforcement learning algorithm. Given a task instruction $q$, the policy $\pi_\theta$ generates a group of $G$ trajectories $\{\tau_i\}_{i=1}^G$. Each trajectory $\tau_i = (s_{i,0}, a_{i,0}, \ldots, s_{i,T_i}, a_{i,T_i})$ consists of $T_i$ timesteps. The GRPO loss is defined as

$$\mathcal{L}_{\text{GRPO}}(\theta) = -\frac{1}{\sum_{i=1}^G T_i} \sum_{i=1}^G \sum_{t=1}^{T_i} \min\Big( r_{i,t}(\theta)\,\hat{A}_{i,t},\ \text{clip}\big(r_{i,t}(\theta),\, 1-\epsilon,\, 1+\epsilon\big)\,\hat{A}_{i,t}\Big), \quad (1)$$

where

$$r_{i,t}(\theta) = \frac{\pi_\theta(a_{i,t} \mid s_{i,t}, q)}{\pi_{\theta_{\text{old}}}(a_{i,t} \mid s_{i,t}, q)}. \quad (2)$$

Since only a binary terminal reward $R(\tau_i) \in \{0, 1\}$ is available for each multi-turn trajectory, we extend GRPO by computing a normalized trajectory-level advantage for each $\tau_i$ using the group mean $\mu$ and standard deviation $\sigma$, and broadcasting it uniformly to all steps in the trajectory:

$$\hat{A}_{\tau_i} = \frac{R(\tau_i) - \mu}{\sigma}, \qquad \hat{A}_{i,t} = \hat{A}_{\tau_i}, \ \ \forall t. \quad (3)$$

In practice, since the VLM generates actions as token sequences, we compute log-probability ratios at the token level and apply GRPO by averaging across tokens within each timestep. For simplicity, we adopt GRPO with KL regularization, omit entropy bonuses, and rely solely on trajectory-level rewards without customized reward shaping. Following Lu et al. (2025), we employ curriculum learning, beginning with easy tasks, then progressing to easy and medium, and finally all tasks.

## 4.2 SCALABLE ROLLOUT COLLECTION

A scalable RL system is essential for large-scale mobile agent training. Since GRPO requires multiple rollouts, which dominate the time consumption of each training step if collected sequentially (see

Figure 7 (left)), parallel collection across environments is critical for efficiency. A simple solution is to use multiprocessing for Android environments, but it encounters scalability limitations:

- **Failure coupling:** Each Android environment operates through an emulator, which imposes substantial CPU and memory overhead. Without resource isolation, multiple processes compete for system resources, leading to instability such as freezes, delays, and crashes. In parallel setups, one failed environment can disrupt others and terminate rollout collection. In addition, because rollout collection is coupled with policy updates, one crash can halt the entire training process.

- **Synchronous rollout barrier**: In practice, the execution time of each environment varies across rollout steps. Some actions, such as text input, complete quickly, whereas others, such as scrolling, are slower. The naive implementation introduces a synchronization barrier that waits for all environments to finish before the VLM policy produces the next batch of outputs. As a result, faster environments idle and the GPUs are underutilized.

To address these limitations, we build a reliable and scalable rollout collection system as in Figure 2:

- **Containerized infrastructure:** To ensure resource isolation and fault tolerance, each Android environment is encapsulated in a Docker container built from a standardized image containing both the Android emulator and a server interface. All containers are assigned identical CPU and memory quotas to prevent stragglers. This design decouples environment execution from policy updates, as each container communicates with the VLM policy model through its server interface via HTTP. During rollout, the agent's outputs (thoughts and actions) or control commands (e.g., reset) are transmitted to the assigned container, where actions are executed and the resulting reward and next-step screenshot are returned.

- **Asynchronous rollouts:** Each environment progresses independently; once an environment execution is completed, the resulting screenshot and reward are immediately returned to the agent to generate the next thoughts and action, rather than waiting for all environments to finish. This eliminates global synchronization, pipelines environment execution with action generation, and improves GPU utilization and throughput.

- **Error recovery:** Each Android emulator is monitored within its Docker container and automatically reinitiated upon failure. A history log records rollout progress, allowing a restarted emulator to resume from the last completed step rather than restarting the entire rollout, thereby accelerating collection and improving reliability.

## 5 EXPERIMENTS

We evaluate how online RL enhances mobile agents' decision-making, generalization in AndroidWorld-Generalization, and RL training system efficiency, addressing four key questions:

1. **Q1: Can RL improve the decision-making capabilities of mobile agents?** We compare RL against supervised fine-tuning in Unseen Instance regime and track performance improvements across evaluation subsets defined by task type and difficulty level.

2. **Q2: Can RL generalize to increasingly challenging unseen scenarios?** We evaluate whether the three regimes in AndroidWorld-Generalization pose progressively greater generalization challenges, and analyze skill transfer in Unseen Template via a case study.

3. **Q3: Can few-shot adaptation at test-time improve performance on unseen apps?** We evaluate whether fine-tuning mobile agents using limited interaction data collected during deployment in Unseen App can improve performance.

4. **Q4: Can the proposed rollout collection system accelerate RL training?** We ablate the asynchronous design to quantify speedup over the naïve AndroidWorld implementation.

**Environment and Training Setting.** We follow AndroidWorld and use an Android 13 emulator (API level 33) with 20 pre-installed apps. We use UI-Tars-7B-SFT as the base model, modifying its prompt template by adding "answer" to the action space to support information-retrieval tasks. Each task instance generates eight rollouts capped at 20 steps due to GPU memory constraints. We collect rollouts with a sampling temperature of $\tau = 1.0$ and adopt Adam optimizer (Kingma, 2014) with a fixed learning rate of $1 \times 10^{-6}$ and a 100-step linear warmup. Each experiment uses 16 parallel environments and two NVIDIA H100-80GB GPUs. More details are provided in Appendix I & J.

**Baselines.** We compare our RL-trained screenshot-based agent with (a) state-of-the-art agents built on proprietary VLMs with prompting, (b) supervised fine-tuning on static human or synthetic demonstrations, and (c) recent RL-based methods. Due to resource constraints, we focus on 7B models but also report results for models up to 72B.

## 5.1 RESULTS AND FINDINGS

**Q1: Can RL improve the decision-making capabilities of mobile agents?** Although the Unseen Instance regime trains on 78 templates, we expand the evaluation set to all 116 templates to remain consistent with the original AndroidWorld evaluation, and maintain comparability with prior baselines. Building on the UI-Tars-7B-SFT baseline, our RL method employs a curriculum learning scheme that expands training tasks from easy, to easy + medium, and eventually to to the full task set defined by AndroidWorld's difficulty scores. This approach more than doubles the average success rate, yielding a 26.1% overall improvement (Table 2) with consistent gains across all difficulty levels—28.4% on Easy, 26.8% on Medium, and 17.5% on Hard (Appendix D). Our method also outperforms proprietary prompting-based methods such as GPT-4o and Claude Computer Use, despite using a much smaller open-source 7B model, and even surpasses larger open-source agents such as UI-TARS-72B-SFT. These results highlight the effectiveness of RL post-training in interactive mobile environments. However, most state-of-the-art methods report benchmark results that should be treated as references rather than strict baselines, since their codebases are not publicly released and therefore not directly reproducible. Accordingly, we do not claim state-of-the-art performance; instead, we emphasize our clear improvements over all reproducible baselines.

Table 2: Comparison of methods on AndroidWorld. **"API-Based"** denotes use of proprietary inference APIs. **"Open-Source"** denotes availability of model weights. **"Reproducible"** denotes availability of full codebase.

| Models | API-Based | Open-Source | Reproducible | Average (SR) |
|---|---|---|---|---|
| *Proprietary model* | | | | |
| GPT-4o (Hurst et al., 2024) | ✓ | | ✓ | 34.5 |
| Claude Computer Use (Anthropic, 2024) | ✓ | | ✓ | 27.9 |
| UGround+GPT-4o (Gou et al., 2024) | ✓ | ✓ | ✓ | 44.0 |
| Aria-UI+GPT-4o (Yang et al., 2024) | ✓ | ✓ | ✓ | 44.8 |
| Agent S2 (Agashe et al., 2025) | ✓ | | | 54.3 |
| *32B/72B Models* | | | | |
| Qwen2.5-VL-32B (Bai et al., 2025b) | | ✓ | ✓ | 31.5 |
| MobileGUI-32B (Shi et al., 2025) | | | | 44.8 |
| AGUVIS-72B (Xu et al., 2024) | | | | 26.1 |
| Qwen2.5-VL-72B (Bai et al., 2025b) | | ✓ | ✓ | 35.0 |
| UI-TARS-72B-SFT (Qin et al., 2025) | | ✓ | ✓ | 46.6 |
| MobileUse-72B (Li et al., 2025) | | ✓ | ✓ | 62.9 |
| *2B/7B Models* | | | | |
| AppVLM-3B (Papoudakis et al., 2025) | | | | 37.8 |
| MobileGUI-7B (Shi et al., 2025) | | | | 30.0 |
| UI-TARS-7B-SFT (Qin et al., 2025) | | ✓ | ✓ | 23.0 ± 2.2 |
| Ours-7B w/o curriculum learning | | ✓ | ✓ | **45.1 ± 2.5 (+22.1)** |
| Ours-7B | | ✓ | ✓ | **49.1 ± 8.2 (+26.1)** |

**Learning Dynamics:** Figure 3 (left) shows consistent improvement of average success rate in both training and evaluation with successive policy updates. All test curves are obtained by evaluating saved checkpoints only after the full training has completed, rather than querying the test set during training. This protocol prevents any train–test leakage and allows an assessment of the gap between training performance and generalization. The systematic evaluation across three difficulty levels and two task types in Figure 3 (right) further demonstrates performance gains, particularly for information retrieval with an initial success rate of 0. Training on easy tasks transfers to medium and hard ones, but information retrieval at the hard level remains unsolved.

**Q2: Can RL generalize to increasingly challenging unseen scenarios?** To study generalization in RL, we train and evaluate mobile agents on three unseen regimes in the AndroidWorld-Generalization benchmark: Unseen Instance/Template/App. For fair comparison, we use the same hyperparameters and 500 policy iteration steps across all experiments. As shown in Figure 4, evaluation success rates increase substantially in Unseen Instance (21.8%). In contrast, gains in Unseen Template (15.7%) and Unseen App (8.3%) are noticeably smaller and plateau early despite continued improvements in training success rates, highlighting that generalizing to new templates and applications remains challenging.

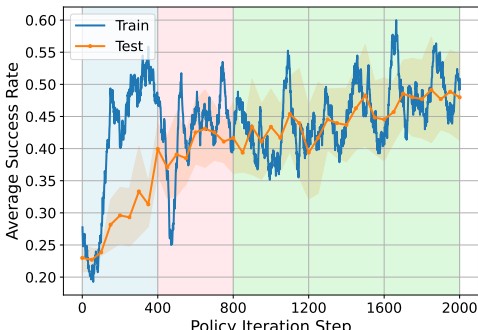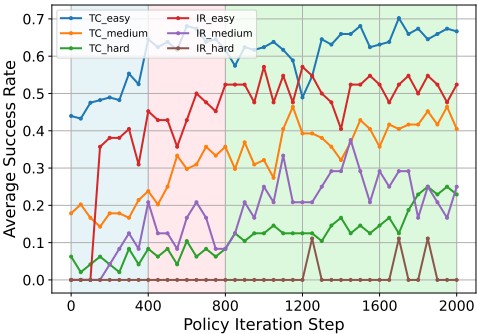

Figure 3: Training dynamics on Unseen Instance with curriculum learning. Colored areas denote curriculum stages: blue (Easy), red (Easy + Medium), green (All). (Left) Average training and evaluation success rates. (Right) Average evaluation success rates by task type (Task Completion, Information Retrieval) and difficulty (Easy, Medium, Hard).

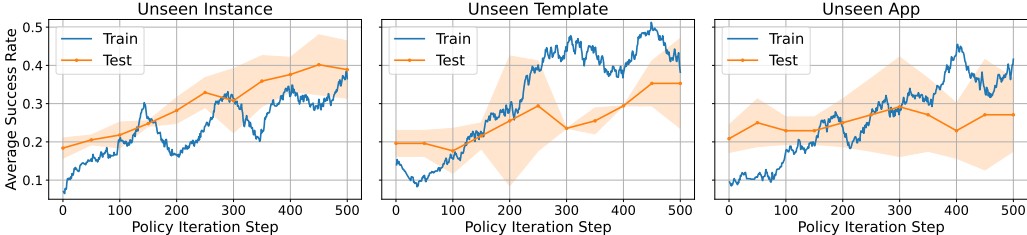

Figure 4: Training dynamics of GRPO across the three unseen regimes. We report training success rates and evaluation success rates with standard deviations.

**Case Study:** To understand generalization in Unseen Template, we analyze templates that failed before RL training but succeeded afterward, revealing the underlying transfer pattern. Although the template itself is unseen, completing it requires leveraging transferable skills from seen templates. For example, an unseen template that requires deleting a food recipe from a list of candidates relies on the skill of identifying the target content. This skill can be transferred from a seen template involving food-name search. Details of the transferable skills are provided in Appendix E.

**PPO.** To demonstrate that our RL training system is algorithm-agnostic, we also evaluate PPO on all three unseen regimes in AndroidWorld-Generalization. We follow a simple PPO baseline with token-level GAE as in (Wang et al., 2025a), where a binary reward is assigned only at the final token of each trajectory and propagated backward to compute token-level advantages. As shown in Figure 5, PPO exhibits the same qualitative trends as GRPO: evaluation success rates improve by 16.7% on Unseen Instance, 9.8% on Unseen Template, and 8.3% on Unseen App.

**Q3: Can few-shot adaptation at test-time improve performance on unseen apps?** While the primary focus of this study is evaluating zero-shot generalization to unseen scenarios, inspired by Beck et al. (2025), we investigate whether simple few-shot fine-tuning at test time can improve performance on the most challenging Unseen App.

To remain consistent with the standard Unseen App regime, we use its 48 instances for testing. For training, we generate 8 non-overlapping instances per unseen app with different random seeds, denoted as `unseen-app-train`. Starting from a model trained for 500 steps on seen apps as the non-adaptation baseline, we allow 50 additional fine-tuning steps on `unseen-app-train` at

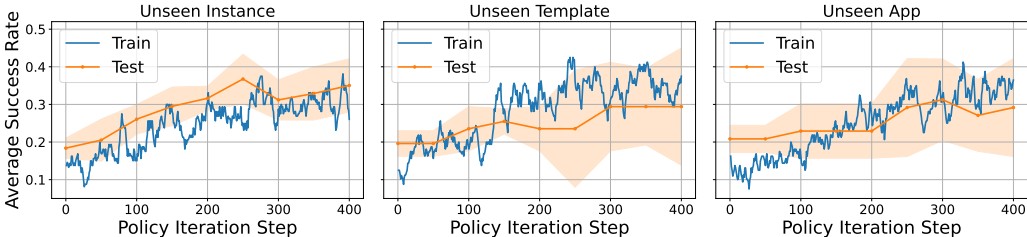

Figure 5: Training dynamics of PPO across the three unseen regimes.

test time using the rule-based reward function, as adaptation stage in practical deployment typically operates on resource-constrained devices and must rely on few-shot data and limited computation.

We propose two adaptation strategies: (i) **All-App**, where the model is fine-tuned on all `unseen-app-train` instances; and (ii) **Per-App**, where separate models are fine-tuned on the 8 instances of each unseen application, enabling stronger personalization. Fig. 6 shows that Per-App adaptation outperforms the non-adapted baseline by 10.4% and All-App adaptation by 6.3%, highlighting the effectiveness of personalized finetuning and underscoring few-shot adaptation as a promising direction for improvement on unseen scenarios.

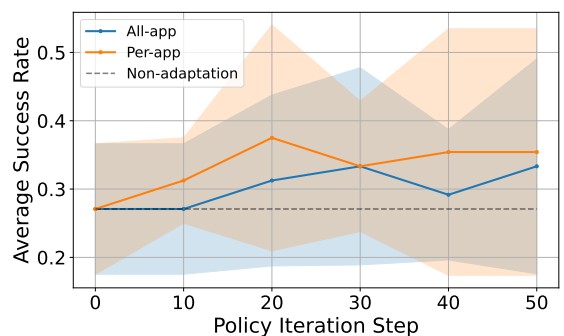

Figure 6: Few-shot adaptation vs. non-adaptation, averaged. over Unseen App test-set.

**Q4: Can the proposed rollout collection system accelerate RL training?** To analyze the impact of parallelization on rollout collection, we profile training with 16 rollouts per training step. While policy update time remains unchanged, rollout collection dominates training. Using 16 environments in parallel reduces collection time by 6.83× compared to a single environment in the sequential setting. To evaluate the effectiveness of our asynchronous design, we measure the average rollout collection time across all evaluation tasks. Without asynchrony, the trainer must wait for all environments to complete, causing the longest rollout in each group to bottleneck progress. This effect amplifies with larger group sizes, leading to a 57.8% slowdown when using 16 environments. Detailed settings are provided in Appendix F.

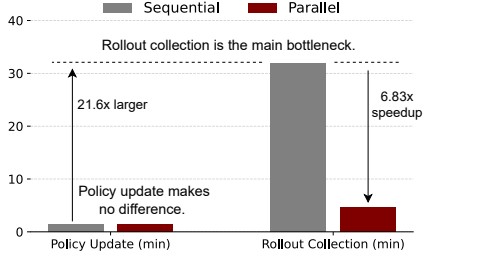 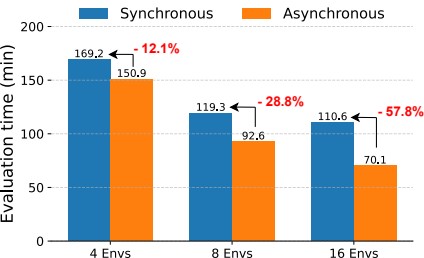

Figure 7: (Left) Time profiling of a policy update vs. the collection of 16 rollouts per training step. (Right) Performance of async vs. sync rollouts with varying environment counts.

## 6 CONCLUSION

In this work, we formulate GUI-based mobile use as a Contextual Markov Decision Process and introduce AndroidWorld-Generalization, a benchmark with three unseen regimes for studying RL generalization in online reinforcement learning. To enable reproducibility, we develop the first fully open-source end-to-end RL framework, integrating GRPO with a scalable rollout system. Experiments show that RL significantly outperforms supervised finetuning but struggles on unseen templates and apps. This work establishes both algorithmic and system foundations for RL-based mobile agents, highlighting to future directions in generalization and few-shot adaptation at test-time.

REPRODUCIBILITY STATEMENT

Our proposed benchmark and end-to-end RL framework for mobile use agents are fully open-sourced. Task instances for all three regimes are publicly released, with detailed generation procedures provided in Appendix C. We additionally release the complete training framework, including the interactive environment and configurations, task sets, prompt templates, agent logic, training hyperparameters, and RL infrastructure, covering both the trainer and the rollout collection system. The anonymized codebase is available at `https://anonymous.4open.science/r/AndroidWorldGeneralization-BE55`.

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

## A    LIMITATIONS

While our benchmark covers three unseen scenarios, its scale is still constrained to 20 applications and 116 templates, which limits both the comprehensiveness of generalization evaluation and the availability of sufficiently diverse training contexts to improve it. Scaling to a larger pool of applications and task templates would not only provide stronger coverage of real-world variability but also offer richer opportunities for agents to acquire transferable skills across heterogeneous tasks. Furthermore, although our rollout collection system is designed with containerized infrastructure to scale to hundreds of environments across multi-node clusters, in practice our current experiments are restricted to a single node due to resource limitations common in academic settings. This restriction prevents us from fully demonstrating the scalability of the system and limits the pace of large-scale training.

## B    LLMS USAGE

ChatGPT5 is used solely as a general-purpose writing assistant. Specifically, we apply a fixed prompt template — "Polish the writing in a concise and academic way" — to improve grammar, clarity, and style of text written by the authors. The LLM did not contribute to research ideation, methodology design, experimental setup, analysis, or result interpretation.

## C    ANDROIDWORLD-GENERALIZATION BECHMARK

**AndroidWorld.**    AndroidWorld is an interactive benchmark built on the Android Emulator, pre-installed with 20 applications and 116 task templates. Each template defines a workflow (e.g., creating a calendar event, deleting a recipe entry) with parameter slots (e.g., event name, date, recipe type) that can be instantiated with different values to generate unlimited task instances via random seeds. Tasks span two categories: *task completion* (e.g., opening an app and completing a form) and *information retrieval* (e.g., extracting a contact name from the address book). Templates are further grouped into three difficulty levels—*easy*, *medium*, and *hard*—based on the number of steps and reasoning complexity required. Task success is evaluated via rule-based scripts. However, AndroidWorld was originally designed for evaluation rather than training, and its codebase only supports sequential execution without parallel rollout collection.

**Train–Test Split on Unseen Instance.**    For evaluation, we follow the AndroidWorld protocol and use all 116 task templates, enabling direct comparison with prior work. For each template, we report mean and standard deviation over three evaluation seeds—seed 30 (the AndroidWorld default) and seeds 7 and 1234—chosen to maximize the number of unique instances. For training, we exclude 38 templates that consistently produce identical task instances regardless of seed, leaving 78 templates. From these, we generate distinct task instances using 16 random seeds $\{1, 2, 3, 4, 5, 6, 8, 9, 12, 123, 12345, 123456, 1234567, 12345678, 123456789, 1234567890\}$, yielding $16 \times 78 = 1248$ instances. To avoid overlap between train and test, we manually remove duplicates, resulting in a final training set of 1149 unique task instances.

**Train-Test split on Unseen Template**    Given the 116 templates in AndroidWorld, we construct the Unseen Template scheme by splitting them into training and evaluation sets according to the following rules. First, we require both seen and unseen templates in the training and evaluation sets to come from overlapping applications. Therefore, we filter out applications that contain only a single template. To further divide training and evaluation, we target a 3:1 ratio while maintaining comparable average difficulty levels (difficulty is defined in the following subsection). Specifically, for each application, if a medium- or hard-level template is assigned to the evaluation set, at least one template of equal or lower difficulty from the same application is retained in the training set. For each template, we then generate distinct task instances using the same automatic generation mechanism with non-overlapping random seeds, as described earlier.

As a result, the filtered dataset contains 14 applications: the training set includes 57 templates (mean difficulty = 1.70) with 836 instances, and the evaluation set includes 18 templates (mean difficulty = 1.72) with 54 instances.

**Train–Test Split on Unseen Apps**  Following the Unseen Instance scheme, 38 templates could not be used for instance generation due to random seed constraints in three applications, leaving 17 applications for the Unseen App regime. Applying the same 3:1 split rule and ensuring similar difficulty levels, we construct the dataset as follows: the training set consists of 62 templates (mean difficulty = 1.68) with 905 instances from 12 applications, while the evaluation set consists of 16 templates (mean difficulty = 1.69) with 48 instances from 5 non-overlapping applications.

**Few-shot adaptation at test-time: Unseen-app Train-Test split**  In the Unseen App scheme, the test set contains 16 task templates across 5 unseen applications: Audio Recorder (1 template), Clock (1), OSMAnd (2), Tasks (4), and Broccoli (8). To enable few-shot adaptation at test time, we construct a corresponding training set from these templates. Specifically, we sample 8, 8, 4, 2, and 1 task instances from the 1, 1, 2, 4, and 8 templates, respectively, using different random seeds. This yields a balanced training set, denoted as `unseen-app-train`, in which each unseen application contains 8 distinct task instances. We manually verified that none of these instances overlap with those in the test set.

**Difficulty calculation**  AndroidWorld includes 116 task templates, each assigned a difficulty level: easy (1), medium (2), or hard (3). To maintain comparable difficulty distributions between training and testing splits across the three unseen regimes (Unseen Instance, Unseen Template, and Unseen App), we ensure that the ratio of templates across difficulty levels is preserved. For example, in the Unseen Template regime, easy, medium, and hard templates are evenly divided between training and testing. Similarly, in the Unseen App regime, we select app combinations such that the proportions of easy, medium, and hard templates in the training apps match those in the held-out test apps. As a result, the average difficulty level between training and testing splits remains similar across all regimes, ensuring that generalization is evaluated under comparable task complexity, reported in Table 1.

**AndroidWorld-Generalization vs. AndroidWorld**  A standard benchmark for agentic mobile tasks typically consists of three components: an interactive environment, a task suite with an associated verifier, and a codebase capable of generating and evaluating rollouts. Along the environment dimension, AndroidWorld and AndroidWorld-Generalization are identical: both use the same 20 Android applications and 116 manually curated task templates. However, the remaining dimensions differ substantially. First, in terms of the task suite, AndroidWorld provides only a single task instance per template (116 total) for evaluation, which limits any robustness assessment. In contrast, AndroidWorld-Generalization uses the automatic task-parameterization mechanism to generate three evaluation instances per template, and leverages the remaining non-overlapping seeds to construct thousands of additional instances that form standardized training sets across three unseen generalization regimes (instance, template, and application). Second, regarding RL support, AndroidWorld was designed solely for evaluation (released in May 2024), before the community's shift toward LLM-driven RL (e.g., DeepSeek-R1, Jan. 2025), and therefore includes no RL-capable training split or interface. AndroidWorld-Generalization explicitly supports the RL paradigm, enabling reproducible RL training and systematic study of RL generalization. Third, with respect to infrastructure, AndroidWorld's evaluation pipeline is fully sequential, making large-scale testing prohibitively slow. AndroidWorld-Generalization introduces a parallelized rollout engine (Section 4), providing up to a $16\times$ speedup (Figure 6) and supporting scalable RL training and evaluation. A comparison is summarized in Table 3.

# D  ADDITIONAL EXPERIMENT RESULTS ON UNSEEN INSTANCE

# E  CASE STUDY ON UNSEEN TEMPLATE

To investigate the underlying reasons for zero-shot transfer in the **Unseen Template** regime, we analyze task instances that succeed after RL training but fail before. Although task templates are disjoint, completing a template typically requires one or a few fundamental skills, which can transfer

---

[1]Prior methds marked with $^*$ are evaluated on a sub-set of AndroidWorld.

Table 3: Comparison between AndroidWorld and AndroidWorld-Generalization.

| | AndroidWorld | AndroidWorld-Generalization |
|---|---|---|
| **Environment** | 116 templates across 20 apps | Same 20 apps and templates (subset splits) |
| **Task Suite** | 116 eval instances (1 seed / template) | Multi-seed eval instances; thousands of non-overlapping train instances |
| **Generalization** | None | Three unseen regimes (instance, template, app) |
| **RL Support** | No; evaluation-only | Yes; standardized train set for RL |
| **Infrastructure** | Sequential rollouts | Parallel rollout collection for scalable evaluation & RL training |

Table 4: AndroidWorld evaluation performance for Mobile Agents. [1]

| Models | Easy (SR) | Medium (SR) | Hard (SR) | Average (SR) |
|---|---|---|---|---|
| *Close-source Models* | | | | |
| GPT-4o (Hurst et al., 2024) | - | - | - | 34.5 |
| Claude Computer Use (Anthropic, 2024) | - | - | - | 27.9 |
| UGround+GPT-4o (Gou et al., 2024) | - | - | - | 44.0 |
| Aria-UI+GPT-4o (Yang et al., 2024) | - | - | - | 44.8 |
| Agent S2 (Agashe et al., 2025) | - | - | - | 54.3 |
| *Open-source 32B/72B Models* | | | | |
| Qwen2.5-VL-32B (Bai et al., 2025b) | - | - | - | 31.5 |
| MobileGUI-32B (Shi et al., 2025) | - | - | - | 44.8 |
| AGUVIS-72B (Xu et al., 2024) | - | - | - | 26.1 |
| Qwen2.5-VL-72B (Bai et al., 2025b) | - | - | - | 35.0 |
| UI-TARS-72B-SFT (Qin et al., 2025) | - | - | - | 46.6 |
| MobileUse-72B (Li et al., 2025) | 83.6 | 47.2 | 26.3 | 62.9 |
| *Open-source 2B/7B Models* | | | | |
| AppVLM-3B[*] (Papoudakis et al., 2025) | 57.9 | 27.4 | 8.3 | 37.8 |
| MobileGUI-7B (Shi et al., 2025) | - | - | - | 30.0 |
| UI-Tars-7B-SFT (Qin et al., 2025) | 33.9 ± 5.3 | 13.9 ± 2.8 | 5.3 ± 0.0 | 23.0 ± 2.2 |
| Ours-7B w/o curriculum learning | 62.8 ± 0.9 | 31.5 ± 8 | 14.0 ± 3 | **45.1 ± 2.5** (**+22.1**) |
| Ours-7B | 62.3 ± 5.7 | 40.7 ± 11.6 | 22.8 ± 13.2 | **49.1 ± 8.2** (**+26.1**) |

from seen to unseen templates. This skill sharing explains why policies trained on certain templates can generalize to non-overlapping ones. We highlight task instances from all five unseen templates that achieve success after RL training, identify the transferable skills they require, and trace these skills back to corresponding seen templates that provided them during training, as summarized below.

## F ABLATION STUDY ON ROLLOUT COLLECTION SYSTEM

To highlight the importance of parallel rollout collection and system optimization, we profile the time of a single policy update iteration, which consists of a policy parameter update and the collection of 16 rollouts executed sequentially and in parallel. For this analysis, we select a task instance with 15 steps, reflecting the average length of training tasks, to generate the rollouts, and repeat the experiment three times for robustness. To ensure a fair comparison, we restart the server before each run and ensure that no other processes occupy GPU or CPU resources. The results show that the update time of the policy parameters remains identical between both settings, while rollout collection occupies more than 95% of the total runtime in the sequential case and almost 75% in the parallel case. These findings demonstrate that optimizing rollout collection is essential to alleviate the training bottleneck and accelerate the overall process.

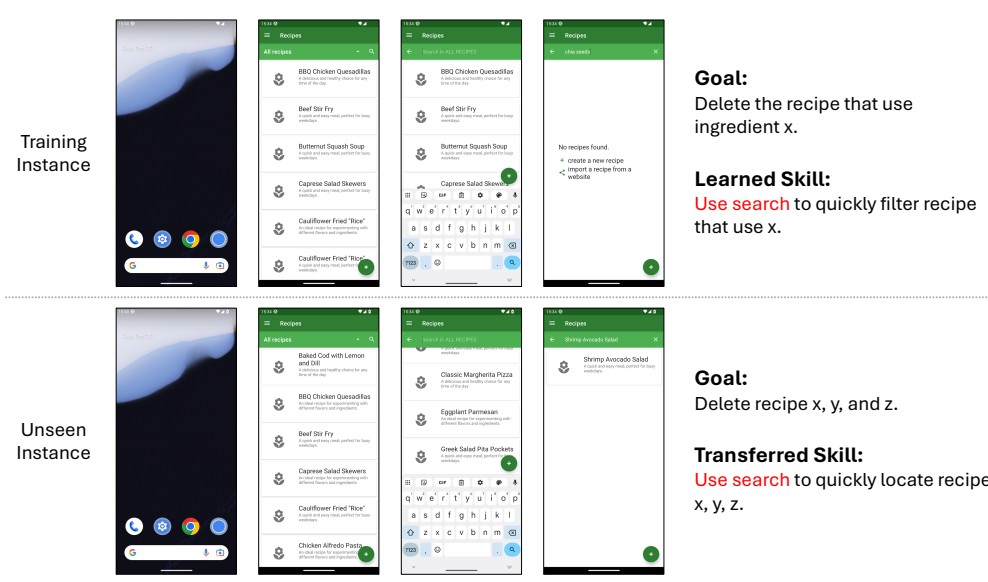

Figure 8: Case study 1.

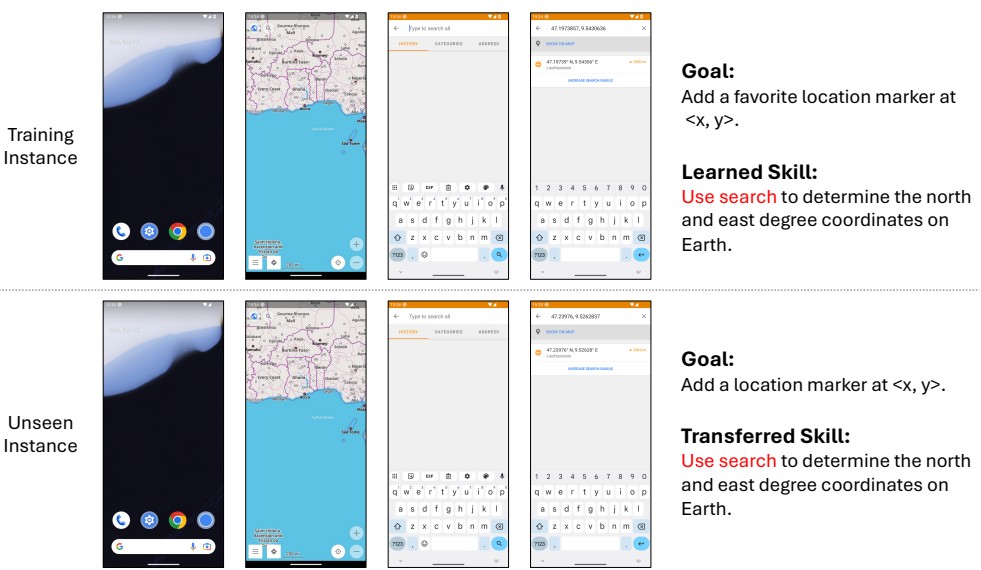

Figure 9: Case study 2.

In our asynchronous design, we adopt a first-come, first-serve strategy: the VLM generates an action as soon as any environment returns its observation, rather than waiting for all environments to complete a full batch at each rollout step. This allows more actions to be executed concurrently and reduces GPU idle time. To evaluate the effectiveness of this design, we ablate the asynchronous mechanism by disabling it and profile the runtime required to complete all 116 task instances in original AndroidWorld evaluation. Varying the number of environments from 4 to 16, we observe that the asynchronous design yields greater benefits as the number of environments increases, since synchronous models are increasingly delayed by the slowest straggler.

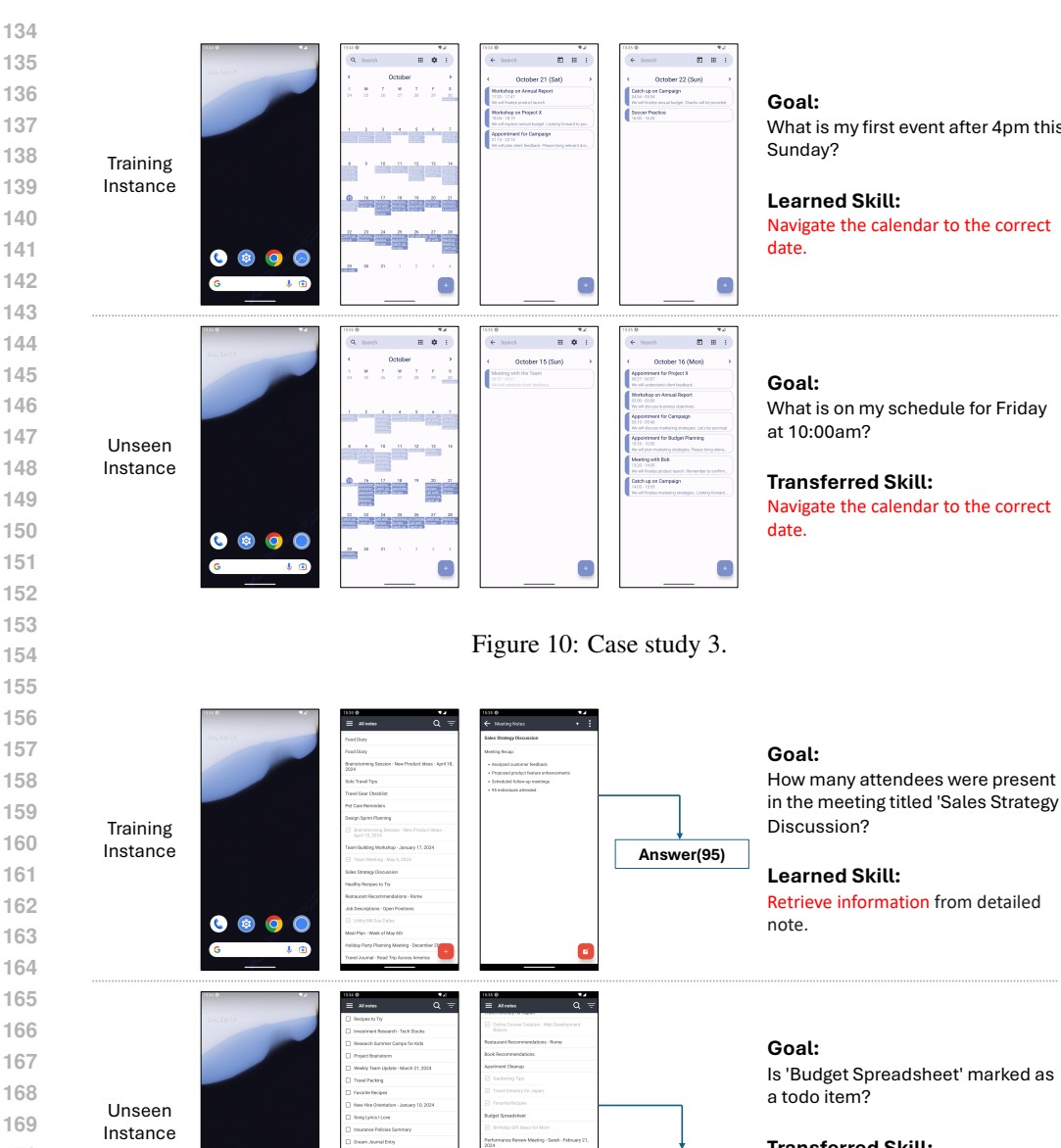

Figure 10: Case study 3.

Figure 11: Case study 4.

## G    REWARD FUNCTIONS: RULE-BASED SCRIPTS VS. LLM-AS-JUDGE

To assess the reliability of reward functions, we compare rule-based scripts with an LLM-as-judge setup by training the same mobile agent using Gemini-2.5-Pro as the reward provider. All training hyper-parameters are kept identical, with the only change being the trajectory reward source (rule-based vs. Gemini). During training, Gemini receives the task description and interaction history (screenshots and actions) as input and produces a binary success score via chain-of-thought reasoning. To mitigate hallucinations, the agent's internal thoughts are excluded from the interaction history. The full prompt is provided below.

```
SYSTEM_PROMPT = """You are an expert evaluator. Your job is
to determine whether the assigned task has been successfully
completed. Base your judgment strictly on visible, verifiable
```

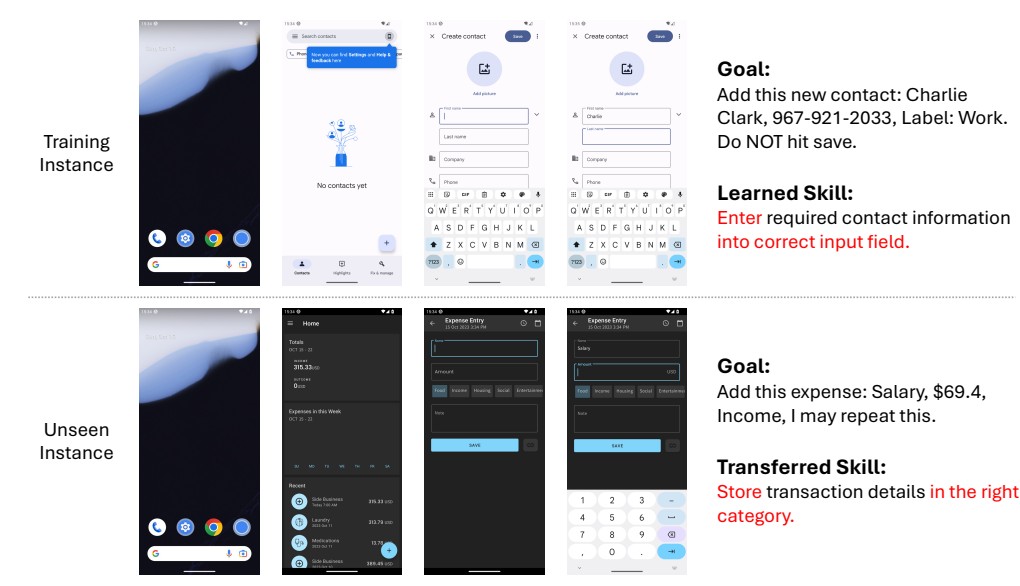

Figure 12: Case study 5.

evidence from the screenshots and action history. Be concise,
objective, and avoid making assumptions beyond what is shown."""

COT_PROMPT = """Task: {task}

Respond using this format:
Thinking: <your concise thought and reasoning process>.
Status: success or failure

Guidelines:
- Include only the two fields above.
- For information retrieval tasks (typically where
  the task requires the agent to provide an answer),
  confirm that the content in the 'answer' action is
  accurate and visibly supported by the screenshots.
- For all other tasks, verify that the task has
  been completed based on clear and observable evidence."""

To simplify the experiment, we use the Easy task subset of the Unseen Instance regime, consisting of 553 task instances across 38 templates and 14 apps. Despite differing reward functions during training, evaluation is always performed with the same rule-based script. As shown in Figure 13, the agent trained with the rule-based script achieves a 26.22% increase in success rate after 400 policy iterations, whereas the agent trained with Gemini reward achieves only an 11.48% improvement, primarily due to false positive signals produced by Gemini. Specifically, Figure 14 shows that Gemini's reward predictions are consistently higher than rule-based script's reward feedbacks, with the gap widening from 10% to 20% as training progresses. Consequently, the model is optimized in a misleading direction, highlighting the importance of reliable rule-based reward functions.

## H  AGENT DESIGN

**Action space.**  The mobile agent interacts with the environment through a set of actions, summarized in Table 5. Together, they form a flexible action space that supports both task execution (e.g., completing forms, navigating menus) and information retrieval (e.g., entering queries). This design balances simplicity with expressiveness, ensuring that the agent can operate across diverse applications, task templates and task instances in AndroidWorld.

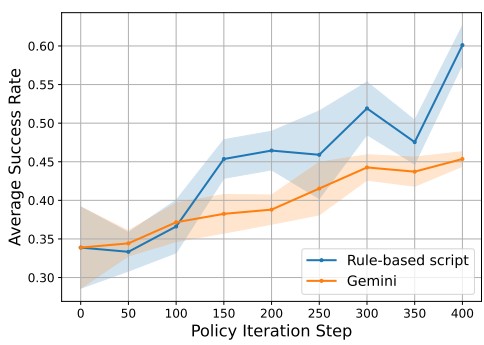 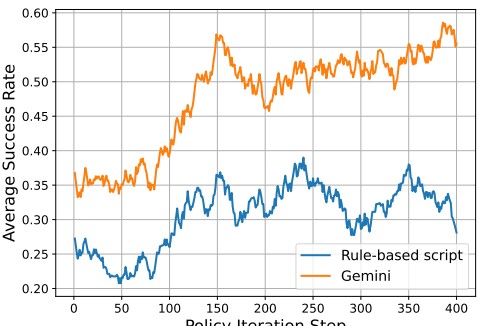

Figure 13: Evaluation success rates using a rule-based script versus Gemini as reward functions. The rule-based script proves more effective at the same number of policy iteration steps.

Figure 14: Training success rates using a rule-based script versus Gemini as reward functions. The rule-based script provides reliable rewards, whereas Gemini assigns higher rewards to the same trajectories, leading to false positives and potentially misleading optimization.

**Prompt Template.** We employ a VLM as the policy model to generate *thoughts* followed by a predicted *action*, conditioned on the current screenshot and the interaction history. To standardize the model's input–output format, we adopt a structured prompt template that specifies how task instructions, screenshots, and histories are provided as input, and how the model should output its reasoning process and corresponding action, as follow

```
UITARS_USR_PROMPT_THOUGHT = """You are a GUI agent. You are given a task
    and your action history, with screenshots. You need to perform the
    next action to complete the task.

## Output Format
```
Thought: ...
Action: ...
```

## Action Space
click(start_box='<|box_start|>(x1,y1)<|box_end|>')
long_press(start_box='<|box_start|>(x1,y1)<|box_end|>', time='')
type(content='') # If you want to submit your input, use "\\n" at the
    end of `content`.
scroll(start_box='<|box_start|>(x1,y1)<|box_end|>',
    end_box='<|box_start|>(x3,y3)<|box_end|>')
press_home()
press_back()
open_app(content='') # Open an app specified by `content`.
finished(content='') # Submit the task regardless of whether it succeeds
    or fails.
answer(content='') # Answer user's question.

## Note
- Use English in `Thought` and `Action` part.
- Write a small plan and finally summarize your next action (with its
    target element) in one sentence in `Thought` part.

## User Instruction
{instruction}
"""
```

Table 5: Action space of mobile use agents in AndroidWorld.

| Action Type | Description |
| --- | --- |
| Click $(x, y)$ | Tap at screen coordinate $(x, y)$ |
| Long press $(x, y)$ | Long press at screen coordinate $(x, y)$ |
| Type (text) | Enter variable-length natural-language text |
| Scroll $(d)$ | Scroll in direction $d \in \{$up, down, left, right$\}$ |
| Navigate home () | Navigate Back to home screen |
| Navigate back () | Navigate Back to previous screen |
| Open (app) | Launch the specified application by name |
| Finish (text) | Report termination of the task |
| Answer (text) | Answer user's question (for Information Retrieval Tasks) |

## I  INTERACTIVE ENVIRONMENT

We follow the AndroidWorld setup procedure to construct our mobile emulator environment on a headless Linux server. The emulator is configured with Android API level 33 (Tiramisu), a resolution of 2400×1080, 16 GB RAM, 6 GB disk space, and 6 virtual CPU cores. All required apps are preinstalled, and their initial launch states are manually verified for correctness. A clean emulator snapshot is saved to eliminate repeated setup overhead during parallel execution. To accommodate action execution, we enforce a fixed delay of 3 seconds and extend it dynamically up to 6 seconds until screen stabilization, determined by comparing accessibility trees across consecutive frames. In contrast, the communication overhead between the Docker server and agents is negligible relative to action latency.

## J  TRAINING DETAILS

We use **UI-Tars-7B-SFT** as the base model. The mobile agent prompt is adapted from the original UI-Tars codebase with minimal modification, adding the required *answer* action to handle *information retrieval* tasks. At each training step, 2 task instances are sampled, and 8 rollout trajectories are collected per instance. Full trajectories are used for weight updates, with trajectory length capped at 20 steps in addition to AndroidWorld task-specific limits to accommodate GPU memory constraints. Under this configuration, each experiment uses 16 virtual environments for trajectory collection and 2 NVIDIA H100 80GB GPUs for action generation and weight updates. All experiments are conducted on a 224-core server with 8 H100 GPUs, supporting up to 3 concurrent training runs (48 Docker containers across 6 GPUs) without latency degradation; performance deteriorates significantly beyond this scale. Training is performed with a maximum learning rate of $1 \times 10^{-6}$, linearly warmed up over the first 5% of steps, and a KL-divergence coefficient of 0.05. Input images are resized to 1120×504, a sampling temperature of 1 is applied, and the main experiment runs for 2000 training steps, requiring approximately 480 GPU hours.

