# OpenReview forum: "Generalization in Online Reinforcement Learning for Mobile Agents"
_ICLR.cc/2026/Conference — Submitted to ICLR 2026_

### Official Review · Reviewer_yrmE · 2025-10-29

**Soundness:** 3
**Presentation:** 3
**Contribution:** 3
**Rating:** 6
**Confidence:** 4

**Summary:**

This paper investigates generalization in online reinforcement learning for GUI-based mobile agents. The authors formalize the problem as a Contextual Markov Decision Process (CMDP) and introduce AndroidWorld-Generalization, a benchmark with three progressive difficulty regimes: Unseen Instance, Unseen Template, and Unseen Application. They develop a fully open-source end-to-end RL framework integrating Group Relative Policy Optimization (GRPO) with a scalable rollout collection system featuring containerized infrastructure, asynchronous execution, and error recovery. Experiments show that RL enables a 7B-parameter VLM agent to achieve good improvement on unseen instances over supervised fine-tuning baselines, but limited gains on unseen templates and apps.

**Strengths:**

- Novel benchmark design: **Generalization is a big issue and bottleneck under this topic.** The three-tiered generalization benchmark (Unseen Instance/Template/App) provides a systematic framework for evaluating zero-shot transfer, addressing a significant gap in existing mobile agent research

- Practical system design: The scalable rollout collection system with Docker containerization, asynchronous execution, and error recovery addresses real engineering challenges in RL for mobile environments

- Insightful case studies: The analysis of transferable skills in Unseen Template provides a valuable understanding of what enables generalization

**Weaknesses:**

- Limited scale: The benchmark covers only 20 applications and 116 templates, which the authors acknowledge constrains generalization evaluation and training diversity

- Poor generalization to harder regimes: The dramatic performance drop on Unseen Template (15.7%) and especially Unseen App (8.3%) suggests fundamental limitations that aren't adequately addressed

- Few-shot adaptation is underdeveloped: The test-time adaptation experiments (Section 5.1, Q3) are preliminary and don't explore important questions like optimal adaptation strategies, data efficiency, or when to apply adaptation

**Questions:**

- Why GRPO specifically? What motivated choosing GRPO over other online RL algorithms like PPO or actor-critic methods? Have you experimented with alternatives?

- Trajectory-level advantages: In Equation 3, you broadcast the trajectory-level advantage uniformly to all timesteps. How does this compare to using per-step advantages or other credit assignment strategies?

- Curriculum learning: What criteria determine progression through curriculum stages (Easy → Easy+Medium → All)? Is this transition automated or manual?

- Adaptation strategy: Why only 50 fine-tuning steps for adaptation? How sensitive is performance to this choice?

- Per-app vs. All-app: The Per-App adaptation outperforms All-App by 6.3%. Does this suggest fundamental limits to cross-app generalization, or could All-App improve with more data?

---

> ### Author Response · Authors · 2025-11-26
>
> Thank you for taking the time to review our paper and provide insightful feedback. To help the reviewer better follow our responses, we reorganize the order of the comments and address each point in turn. In the following, **W** refers to a reported weakness (e.g., W1), and **Q** refers to a specific question (e.g., Q1). We now respond to all concerns below.
>
> > W1: Limited scale: The benchmark covers only 20 applications and 116 templates, which the authors acknowledge constrains generalization evaluation and training diversity
>
> Thank you for raising this point. We agree that the current scale of AndroidWorld (20 apps, 116 templates) limits the breadth of evaluation, and we explicitly acknowledge this in the limitation section (Line 918). However, this limitation stems from the underlying environment itself, not from our contributions in benchmark design for generalization or an RL training system for mobile agents. Expanding GUI-agent benchmarks to hundreds of tasks and a larger set of apps with reliable evaluation protocols is an active, community-wide challenge[1,2,3] and addressing that broader scaling problem falls outside the scope of our work.
>
> Our benchmark and codebase will be fully open-sourced upon acceptance. As richer task-generation methods and larger app suites emerge, our framework is designed to integrate them seamlessly, enabling future scaling without modifying the core methodology.
>
> [1] OS-Genesis: Automating GUI Agent Trajectory Construction via Reverse Task Synthesis. ACL 2025
>
> [2] Scaling Web Agent Training through Automatic Data Generation and Fine-grained Evaluation. COLM 2025
>
> [3] Scaling Synthetic Task Generation for Agents via Exploration. Apple. Sep 2025
>
> > W2: Poor generalization to harder regimes: The dramatic performance drop on Unseen Template (15.7%) and especially Unseen App (8.3%) suggests fundamental limitations that aren't adequately addressed
>
> Thank you for the observation. We agree that performance drops substantially in the Unseen Template and Unseen App regimes, and this reflects the fundamental challenges of RL generalization in mobile agent. Importantly, these findings arise because of our proposed benchmark: prior to our work, there was no principled way to measure generalization across these regimes, so these limitations could not even be identified.
>
> Rather than being a weakness of our approach, the poor generalization revealed by our benchmark demonstrates the need for exactly the kind of standard benchmark and RL training system we propose—a gap that has been largely unexplored in the literature, as noted in the abstract. By providing the first foundation that exposes these challenges in a reproducible way, our work enables and motivates future research to develop more advanced RL methods to address these limitations.
>
> > W3: Few-shot adaptation is underdeveloped: The test-time adaptation experiments (Section 5.1, Q3) are preliminary and don't explore important questions like optimal adaptation strategies, data efficiency, or when to apply adaptation
>
> Thank you for recognizing our preliminary results on test-time adaptation in Section 5.1 (Q3). However, the goal of these experiments is not to propose a full adaptation algorithm, but to demonstrate that even two simple test-time adaptation strategies can already improve performance on the challenging Unseen App regime. This provides early empirical evidence that test-time data can be a promising direction to address the generalization gaps revealed by our benchmark.
>
> As outlined in the Introduction (Line 111), the three contributions of this work are: (1) the benchmark, (2) the system that enables RL training for mobile agents, and (3) the findings revealed by these tools—specifically, the limited generalization to unseen templates and apps(W2), and the identification of few-shot test-time adaptation as a promising direction(W3). These insights are meant to motivate future algorithmic research, rather than propose new adaptation or RL algorithms within this work. Therefore, fully exploring optimal adaptation strategies lies beyond the scope of our paper.

---

> ### Author Response · Authors · 2025-11-26
>
> > Q1: Why GRPO specifically? What motivated choosing GRPO over other online RL algorithms like PPO or actor-critic methods? Have you experimented with alternatives
>
> Thank you for raising this question. Our software framework is intentionally designed to be algorithm-agnostic. Because it builds on VeRL [4] as the RL trainer backend, all RL algorithms implemented in VeRL—including PPO, GSPO, RLOO, DAPO, and others—can be integrated into our system without any modification. GRPO was chosen simply as a representative and widely adopted LLM-RL baseline used in recent reasoning and agentic tasks.
>
> To address the reviewer’s concern, we added **PPO results** in Section 5.1 (Q2) of the revised paper, highlighted in blue. PPO and GRPO are the two most commonly used baseline RL algorithms in current LLM-RL work, and adding PPO demonstrates that our framework supports multiple RL algorithms and that our contributions are not tied to GRPO. Our goal is to provide the infrastructure and benchmark, not to advocate for a specific RL algorithm.
>
> [4] HybridFlow: A Flexible and Efficient RLHF Framework. EuroSys 2025
>
> > Q2: Trajectory-level advantages: In Equation 3, you broadcast the trajectory-level advantage uniformly to all timesteps. How does this compare to using per-step advantages or other credit assignment strategies?
>
> We adopt the simple strategy of broadcasting the trajectory-level advantage uniformly to all timesteps for two reasons: (1) It has been used in prior work [5] and in the standard GRPO baseline in [6], and (2) advanced credit-assignment strategies are not the focus of this paper. Recent NeurIPS 2025 works have explored more expressive alternatives—GiGPO [6], which adds a two-level trajectory/step advantage structure for GRPO, and Bi-Level GAE [7], which introduces step- and token-level advantages for PPO. These approaches outperform their respective baselines in other agentic settings but have not been evaluated in mobile environments. Since both our system and these methods are implemented using VeRL[4] as the backend, they can be readily incorporated into our open-source training pipeline after the paper is accepted and the codebase is released, enabling evaluation of whether finer-grained credit assignment improves mobile-agent performance.
>
> [5] ARPO: End-to-End Policy Optimization for GUI Agents with Experience Replay. May 2025
>
> [6] Group-in-Group Policy Optimization for LLM Agent Training. NeurIPS 2025
>
> [7] VAGEN: Reinforcing World Model Reasoning for Multi-Turn VLM Agents. NeurIPS 2025
>
> > Q3: Curriculum learning: What criteria determine progression through curriculum stages (Easy → Easy+Medium → All)? Is this transition automated or manual?
>
> The difficulty levels are defined by the original AndroidWorld paper, where six experienced annotators rated each template by difficulty, expected duration, and task category. Our curriculum follows a simple, manually specified progression: we begin with easy tasks, expand to the combination of easy and medium tasks, and finally train on the full task set. This transition is not automated. We have added a brief clarification of this curriculum in the revised paper (Line 389). Exploring more sophisticated or adaptive curriculum strategies is an interesting direction for future work, but it is beyond the scope of this paper, whose primary objective is to establish and analyze an RL generalization benchmark rather than to optimize curriculum design.
>
> > Q4: Adaptation strategy: Why only 50 fine-tuning steps for adaptation? How sensitive is performance to this choice?
>
> The choice of a 50-step test-time adaptation reflects practical deployment constraints: while mobile agents are trained in the cloud without computational limits, test-time adaptation typically runs on resource-constrained devices such as smartphones, where compute and memory are limited. Under these conditions, a small number of adaptation steps is realistic and appropriate.
>
> The 50-step budget is therefore a practical default rather than a fixed requirement and can be adjusted depending on deployment needs. In Figure 5, we report performance across 10–50 adaptation steps and observe consistent improvements as the budget increases. We have added an explanation of this design choice in the revised paper in Line 496.

---

> ### Author Response · Authors · 2025-11-26
>
> > Q5: Per-app vs. All-app: The Per-App adaptation outperforms All-App by 6.3%. Does this suggest fundamental limits to cross-app generalization, or could All-App improve with more data?
>
> Per-App and All-App adaptation reflect a trade-off rather than a limitation of either approach. Per-App adaptation personalizes a separate model for each unseen app, which naturally yields higher performance but requires maintaining multiple model copies and performing test-time adaptation for each app. In contrast, All-App uses a single cross-app model, which is more computationally efficient and better aligned with real-world deployment on resource-limited devices.
>
> Our intention is not to suggest that cross-app generalization has fundamental limits; rather, the two results illustrate two complementary and promising research directions. While All-App performance could improve with more adaptation data, scaling data collection and reliable reward verification on unseen apps remains an open challenge.

---

### Official Review · Reviewer_UZoD · 2025-11-01

**Soundness:** 2
**Presentation:** 3
**Contribution:** 2
**Rating:** 2
**Confidence:** 3

**Summary:**

This paper proposes a benchmark, AndroidWorld-Generalization, that evaluates the RL-trained GUI-based mobile agents' capability on generalizing to unseen tasks. They also open-source an RL framework for mobile agent RL training. In this framework, they implement the GRPO algorithm, and leverage asynchronous simulation for better scalability. Finally, they conduct experiments to show the effectiveness of the proposed framework.

**Strengths:**

1. This paper is clearly written and easy to follow
2. This paper studies generalization to new tasks, which is a very important question about algorithm evaluation.
3. The benchmark and training framework is going to be open-sourced

**Weaknesses:**

My main criticism about this paper is that I do not follow the overall logic of the paper. The paper starts with a benchmark, highlighting that it can test the model's performance on unseen tasks. However, they use the environments from AndroidWorld, which limits the novelty. Next, the paper proposes an RL training framework. A framework should facilitate the implementation of many algorithms, whereas the paper only implement GRPO. The paper claims that the the agent trained under this framework outperforms prior works, but I do not see where the improvements come from. See questions for details

**Questions:**

1. To my understanding, for any benchmark that includes multiple tasks, we can group the tasks manually into training set, test set, and validation set. Besides this, what is the novelty of AndroidWorld?
2. How does the RL training framework facilitate the implementation of algorithms besides GRPO? Why are current RL training frameworks not suitable for mobile agent training? How does the scalability of the existing frameworks compare to the framework you proposed?
3. How are the prior works in Table 2 trained? Where is the improvement of this paper coming from over prior works? Training agents using RL and the GRPO algorithm is not new. Are the authors trying to say this paper is the first to use RL on mobile agents?

---

> ### Author Response · Authors · 2025-11-26
>
> Thank you for taking the time to review our paper and provide thoughtful feedback. To clarify our motivation and contributions, we first provide additional background on the emerging field of LLM-based mobile agents, where the rapid pace of development and lack of established conventions can naturally lead to ambiguity at first glance.
>
> Most recent advances in LLM-RL for mobile agents (e.g., [1–6]) come from industry labs and are released as technical reports without open-sourced training or evaluation code. This lack of transparency introduces three fundamental challenges in the current landscape. **First**, without access to training pipelines, prior results cannot be reliably reproduced or fairly compared—especially given that performance varies significantly across random seeds, as shown in [7]. **Second**, the absence of standardized train–test splits prevents systematic study of generalization, making it impossible to evaluate unseen-instance, unseen-template, or unseen-app scenarios in a principled way. **Third**, the lack of an open-source RL system for mobile agents poses substantial barriers for future research: integrating a realistic mobile environment into existing LLM-RL frameworks (e.g., VeRL [8] or SkyRL [9]) is non-trivial due to the absence of a unified environment API and the need for reliable, scalable rollout collection (Section 4). Prior to the ICLR 2026 deadline, the community had open-source the codebases for web tasks [10], computer use [11], tool use [12], and software engineering [13], but no RL training system existed for mobile agents, leaving the area fragmented and difficult to build upon.
>
> Motivated by these gaps, **our work focuses on providing the missing foundations rather than proposing a new algorithm**. We introduce **a formal CMDP formulation** of generalization in mobile-agent RL, **a benchmark** with standardized zero-shot evaluation across three unseen regimes, and the **first open-source RL training system** for mobile agents with efficient GRPO-based training and scalable rollout collection. These contributions establish the necessary infrastructure for fair comparison, reproducible experimentation, and future algorithmic innovation in this emerging domain.
>
> Below, we answer the reviewers’ questions and address the concerns in detail. To help the reviewer better follow our responses, we reorganize the order of the comments and address each point in turn. In the following, **W** refers to a reported weakness (e.g., W1), and **Q** refers to a specific question (e.g., Q1). In the revised version, we further emphasize the engineering challenges in the current landscape and clarify that the focus of the paper is not on algorithmic innovation, but on establishing the problem formulation, benchmark, and infrastructure necessary to enable progress in this emerging research field. All updates are highlighted in blue (See Introduction)
>
>
> [1] UI-TARS: Pioneering Automated GUI Interaction with Native Agents. ByteDance. Jan 2025
>
> [2] Appvlm: A lightweight vision language model for online app control. Huawei. Feb 2025
>
> [3] ZeroGUI: Automating Online GUI Learning at Zero Human Cost. Shanghai AI Lab. May 2025
>
> [4] MobileGUI-RL: Advancing Mobile GUI Agent through Reinforcement Learning in Online Environment. Tecent. July 2025
>
> [5] Mobile-R1: Towards Interactive Reinforcement Learning for VLM-Based Mobile Agent via Task-Level Rewards. Alibaba. Aug 2025
>
> [6] Agent s2: A compositional generalist-specialist framework for computer use agents. Simular AI (startup). COLM 2025
>
> [7] AndroidWorld: A Dynamic Benchmarking Environment for Autonomous Agents. ICLR 2025
>
> [8] HybridFlow: A Flexible and Efficient RLHF Framework. EuroSys 2025
>
> [9] SkyRL-v0: Train Real-World Long-Horizon Agents via Reinforcement Learning. May 2025
>
> [10] WebRL: Training LLM Web Agents via Self-Evolving Online Curriculum Reinforcement Learning. ICLR 2025
>
> [11] ARPO: End-to-End Policy Optimization for GUI Agents with Experience Replay. May 2025
>
> [12] VerlTool: Towards Holistic Agentic Reinforcement Learning with Tool Use. Sep 2025
>
> [13] SWE-RL: Advancing LLM Reasoning via Reinforcement Learning on Open Software Evolution. NeurIPS 2025

---

> ### Author Response · Authors · 2025-11-26
>
> > W1.1: Since AndroidWorld provides environments that already exist, what is the novelty of the proposed AndroidWorld-Generalization compared with AndroidWorld?
>
> A standard benchmark for agentic tasks typically includes three components: an interactive environment, a task suite with an associated verifier (LLM-as-judge or rule-based), and a codebase capable of generating and evaluating rollouts for agent–environment interaction. In terms of the interactive environment, AndroidWorld and our proposed AndroidWorld-Generalization are identical, as both use the same 20 Android apps and 116 manually curated task templates. However, the remaining dimensions differ substantially. First, in the task suite, AndroidWorld provides only one task instance per template (116 total) for evaluation, which limits any assessment of robustness. In contrast, AndroidWorld-Generalization uses an automatic task-parameterization mechanism to generate three task instances per template using three random seeds for robustness evaluation, and leverages the remaining non-overlapping seeds to create thousands of additional instances that form training sets supporting standardized train–test splits across three unseen regimes (instance, template, and app). Second, regarding RL support, AndroidWorld was designed solely for evaluation (May 2024) prior to the community’s shift toward LLM-RL (DeepSeek-R1, Jan 2025) and therefore offers no support for RL; AndroidWorld-Generalization explicitly supports the RL paradigm and enables systematic study of RL generalization. Third, in terms of codebase/infrastructure, AndroidWorld’s evaluation pipeline is fully sequential, making large-scale testing impractically slow, whereas AndroidWorld-Generalization introduces a parallelized rollout system (Section 4) that achieves a 16× speedup, as shown in Figure 6, and supports scalable RL training. The comparison is summarized in the following table and updated in Section C of the revised paper.
>
> | **Dimension**                  | **AndroidWorld**                         | **AndroidWorld-Generalization (Ours)**                                                                     |
> | ------------------------------ | ---------------------------------------- | ---------------------------------------------------------------------------------------------------------- |
> | **Interactive Environment** | Same 20 apps and 116 task templates      | Same environment and templates                                                                             |
> | **Task Suite**        | Test-set: 116 task instances (1 seed per template) | Test-set: Multi-seed evaluation (3 seeds per template); Train-set: thousands of non-overlapping seeds for train/test splits |
> | **Generalization Scenarios**   | None                                     | Unseen-task, unseen-template, unseen-app                                                         |
> | **RL Support**              | No RL paradigm; no training split        | Full RL support with standardized training sets and infra                                      |
> | **Rollout Infrastructure**  | Sequential evaluation only               | Parallelized rollout engine enabling scalable evaluation & RL training                                     |
>
> >Q1: To my understanding, for any benchmark that includes multiple tasks, we can group the tasks manually into training set, test set, and validation set. Besides this, what is the novelty of AndroidWorld?
>
> AndroidWorld is designed solely for evaluation and provides only a test set, with no train or validation set. The reviewer’s suggestion of manually grouping the 116 task templates into train/validation/test sets corresponds only to one of our three proposed unseen regimes—specifically, the unseen-template regime.
>
> In addition, we intentionally do not prescribe a fixed validation split for two reasons. First, most prior influential RL generalization benchmarks [14, 15] provide only train–test splits, as RL model selection differs fundamentally from supervised learning. Second, as shown in [16], validation-set performance in RL is often less reliable for hyperparameter tuning and model selection, and high validation performance does not necessarily translate to improved test-set generalization. Therefore, we leave the choice of validation partitioning to users, allowing them to derive a validation set from the training pool according to their specific algorithmic needs.
>
> [14] Quantifying Generalization in Reinforcement Learning. ICML 2019
>
> [15] Why generalization in RL is difficult: epistemic POMDPs and implicit partial observability. NeurIPS 2021
>
> [16] Empirical design in reinforcement learning. The Journal of Machine Learning Research, Volume 25, Issue 1

---

> ### Author Response · Authors · 2025-11-26
>
> > W1.2 & Q2 the paper proposes an RL training framework. A framework should facilitate the implementation of many algorithms, whereas the paper only implement GRPO. How does the RL training framework facilitate the implementation of algorithms besides GRPO?
>
> Thank you for raising this question. The term *framework* can refer either to a conceptual algorithmic framework or to a software framework. In our paper, we use “framework’’ to denote a software system and codebase for RL on mobile agents. Since our primary contribution is an open-source engineering system rather than a new RL algorithm, we have followed the reviewer’s suggestion and replaced “RL training framework’’ with “**RL training system**’’ throughout the revised manuscript to avoid ambiguity.
>
> We intentionally use GRPO as our baseline because it is simple, widely adopted in LLM-RL, and sufficient for demonstrating the effectiveness of the proposed system. Importantly, the system itself is **algorithm-agnostic**. Built on VeRL [8] as the backend trainer, it can directly support most RL algorithms already implemented in VeRL—including PPO, GSPO, RLOO, DAPO, and others—without any code modification. To further address the reviewer’s concern, we added **PPO results** in Section 5.1 (Q2) of the revised paper to show that the system naturally supports multiple RL algorithms and that our contributions are not tied to GRPO specifically.
>
> > Q2: Why are current RL training frameworks not suitable for mobile agent training? How does the scalability of the existing frameworks compare to the framework you proposed?
>
> Current open-source agentic LLM-RL software frameworks and libraries that use VeRL as the trainer backend—such as SkyRL [9] and rLLM [17]—do not support mobile-agent scenarios. They cover only a limited set of downstream environments (e.g., coding[9], SQL[18], web search[19]), and their architecture designs and APIs have changed significantly in recent months, making integration unstable. More general PyTorch-native frameworks such as TorchForge [20] and OpenEnv [21] were released in late October, after the ICLR submission deadline, and are still in early development without community-verified effectiveness. Consequently, no open-source RL training system existed for large-scale RL on LLM-based mobile agents prior to our work, and integrating mobile environments into rapidly evolving LLM-RL software frameworks would require substantial engineering effort, as detailed in Section 4.
>
> Regarding scalability, a direct comparison to prior works is not feasible because ours work proposes the first system to integrate Android mobile environments into an LLM-RL framework with parallel rollout collection. Existing codebases either do not support mobile environments at all or rely on sequential evaluation pipelines, which cannot scale to the volume of rollouts required for RL.
>
> [17] rLLM: A Framework for Post-Training Language Agents. July 2025
>
> [18] SkyRL-SQL: Multi-turn SQL Data Agents via RL. NeurIPS 2025 Workshop: Multi-Turn Interactions
> in Large Language Models.
>
> [19] Search-R1: Training LLMs to Reason and Leverage Search Engines with Reinforcement Learning. COLM 2025.
>
> [20] Torchforge: A PyTorch native library for scalable RL post-training and agentic development. Oct 2025
>
> [21] OpenEnv: Agentic Execution Environments. Oct 2025

---

> ### Author Response · Authors · 2025-11-26
>
> > W3 & Q3: The paper claims that the the agent trained under this framework outperforms prior works, but I do not see where the improvements come from. How are the prior works in Table 2 trained? Where is the improvement of this paper coming from over prior works?
>
> Most prior methods—particularly the RL-based approaches [2,4,5]—are not open-sourced, and their training datasets and training recipes are not documented. Consequently, it is unclear how evaluation tasks influenced the construction of their training sets, raising concerns about potential leakage. As these methods are not reproducible, a fair comparison is not feasible. Therefore, the numbers reported for prior work should be interpreted only as references rather than strict baselines, and we do not claim state-of-the-art performance over these unreproducible results. This discussion has been incorporated into Section 5.1-Q1 of the revised paper.
>
> To further clarify the comparison with prior work, we update Table 2 to include additional columns—Proprietary API, Uses Open-Source Model, and Reproducible via Open-Source Agent Codebase—to clearly indicate which methods allow strict and fair comparison. For fairness, we only compare against methods that can be reliably reproduced. As discussed in the paper, these fall into two categories: (1) proprietary foundation models such as GPT-4o and Claude, and (2) the SFT baseline UI-TARS. Our RL-trained agent outperforms those reproducible non-RL baselines, showing that **the performance gains stem specifically from applying reinforcement learning** to learn policies that generalize more effectively.
>
> In summary, we do not introduce a new or advanced RL algorithm. Our contribution is the first open-source RL training system that enables GRPO to be used as a reproducible RL baseline for mobile agents, allowing an SFT model to be further improved through RL training.
>
>
> > Q3:  Training agents using RL and the GRPO algorithm is not new. Are the authors trying to say this paper is the first to use RL on mobile agents?
>
> We agree that using GRPO as an RL algorithm for training LLM-based agents is not conceptually new. However, practically implementing a scalable and reliable RL infrastructure for mobile agents is highly non-trivial. Unlike web or code environments, mobile interfaces involve complex GUI dynamics, high-latency interactions, device-state variability. Implementing RL in this setting requires substantial engineering, including (i) designing an appropriate action space, prompt template, and action parser to map open-vocabulary LLM outputs to environment-valid actions, and (ii) building the parallel rollout collection system considering asynchronization and error-handling mechanisms described in Section 4.
>
> To the best of our knowledge, this paper is the first open-source work to successfully train an LLM-based mobile agent using online RL, as prior mobile-agent systems did not provide the necessary RL training infrastructure and standardized train/test splits to support RL generalization. Our contribution is therefore not a new RL algorithm, but the first practical and reproducible RL training system that enables RL to be applied effectively in the mobile-agent setting.

---

### Official Review · Reviewer_pnBC · 2025-11-02

**Soundness:** 3
**Presentation:** 3
**Contribution:** 2
**Rating:** 4
**Confidence:** 5

**Summary:**

This paper focuses on the generalization capabilities of device control agents, and introduces a new benchmark that specifically tests generalization in terms of task instances, templates (tasks), and apps. They also experiment with RL-based fine-tuning and test-time adaptation as approaches for improving generalization. The paper's core contributions are to introduce a new benchmark focused on such generalization, and present experiments using RL and test-time adaptation in this generalization problem. In conducting the RL experiments, they also develop software that supports future work in RL training for device control agents.

**Strengths:**

* The formalization of generalization as CDMP is nice.
* Results demonstrate that, like prior work  has found, RL-based fine-tuning can improve agent performance, even in difficult generalization problems.

**Weaknesses:**

* The intro has a ton of citations in it, these should really only be in the related work unless they are central to the core argument of the paper
* The test sizes are very small. What is prohibiting you from generating even more held-out test data, since it's all generated anyway? Especially because in Section 5.1 you are already expanding it to even more templates for this experiment. Perhaps I am wrong about this estimate because the format of Table 1 is confusing, but computing accuracy over just 69 examples (in unseen app) is not going to result in a statistically significant difference when performance gaps are just ~8%.
* Test data should not be used for analyzing training dynamics, e.g., in Fig 3, 4. This serves as a form of information leakage in the experiment, where we are no longer using the test data to evaluate true generalization of the approach to held-out, IID data.

**Questions:**

* I don't understand how Table 1 is formatted. Why are there training examples for unseen instance/template/app? Shouldn't these be held-out until test only? And why do each of the counts also include instance/template/app breakdowns? In general, I'm confused on what the training/testing setup is like in terms of data available at each stage wrt. generalization dimension being tested.
* How is task difficulty determined? This is coming from AndroidWorld, but how do they define it?

---

> ### Author Response · Authors · 2025-11-26
>
> Thank you for taking the time to review our paper and provide insightful feedback. To help the reviewer better follow our responses, we reorganize the order of the comments and address each point in turn. In the following, **W** refers to a reported weakness (e.g., W1), and **Q** refers to a specific question (e.g., Q1). We now respond to all concerns below.
>
> > Q1. I don't understand how Table 1 is formatted. Why are there training examples for unseen instance/template/app? Shouldn't these be held-out until test only? And why do each of the counts also include instance/template/app breakdowns? In general, I'm confused on what the training/testing setup is like in terms of data available at each stage wrt. generalization dimension being tested.
>
> Thank you for the question. We believe the confusion arises from how Table 1 summarizes the counts of apps, templates, and instances, as well as from the placement of the data-generation details in Appendix Section C due to the 9-page submission limit. In the revision, we moved these details into Section 3.2 and reorganized Table 1 to make the training–testing protocol explicit. Below, we summarize the key clarifications.
>
> **App, Template, and Task Instance in AndroidWorld.**
> AndroidWorld contains 116 task templates across 20 apps. Tasks are organized hierarchically: each app contains several templates, and each template can generate many distinct task instances through parameterization. For example, the Google Calendar app includes templates such as "Create an event {event_name} two weeks from today" and "Delete events on {day_of_week}." By varying the parameter (e.g., event_name) using different random seeds, one template can produce many non-identical task instances.
>
> **How training data is generated.**
> The original AndroidWorld only provides evaluation/testing tasks but offers a parameterization mechanism that allows us to create training data. For a given template, we generate non-overlapping sets of instances using different random seeds—e.g., one seed for training and another for testing.
> * **Unseen Instance**: train and test share the same apps and templates but use disjoint seeds. We use 16 seeds for training and 3 for testing per template, yielding 1149 training instances and 234 test instances from 78 templates across 17 apps.
> * **Unseen Template**: train and test share the same apps but use disjoint templates. For example, within Google Calendar, training may use the task template of "create event" while testing uses the template of "delete event". Using the same 16/3 seed ratio per template results in 836 training and 54 test instances across non-overlapping template sets.
> * **Unseen App**: train and test come from disjoint apps (e.g., train on Calendar, test on Contacts), producing 905 training instances from 12 apps and 48 test instances from 5 apps.
>
> The task-instance generation for each of the three unseen regimes is performed independently.
>
> **Ensuring non-overlap across three generalization dimensions.**
> After generating thousands of task instances, we manually verified that apps, templates, and instance seeds do not overlap between train and test according to the definition of each regime.
>
> **Interpreting Table 1.**
> To avoid ambiguity, the revised Table 1 adds an “overlap’’ column indicating whether apps and templates overlap between train and test. Bold numbers denote the number of task instances used for training, while gray numbers denote the number of apps/templates from which those instances are generated. For example, in the Unseen Instance regime, apps and templates are overlap, but instance seeds are not. In Unseen Template, only apps overlap. In Unseen App, no components overlap.
>
> These clarifications are incorporated into the revised paper (highlighted in blue), and Table 1 has been reformatted accordingly.

---

> ### Author Response · Authors · 2025-11-26
>
> > W2. The test sizes are very small. What is prohibiting you from generating even more held-out test data, since it's all generated anyway? Especially because in Section 5.1 you are already expanding it to even more templates for this experiment. Perhaps I am wrong about this estimate because the format of Table 1 is confusing, but computing accuracy over just 69 examples (in unseen apps) is not going to result in a statistically significant difference when performance gaps are just ~8%.
>
> We agree that larger test sets are desirable. However, increasing the held-out test size is not unconstrained, even with the task generation process as described above. To clarify our design choices in the proposed benchmark, we provide additional context below. Specifically, constructing reliable train–test splits required us to satisfy three strict constraints. (1) We required strict non-overlap between train and test instances across all three regimes. In practice, AndroidWorld’s task parameterization mechanism begins producing overlapping or near-duplicate tasks when too many seeds are used; empirically, 3 seeds for test and 16 seeds for train is the largest configuration that maintains guaranteed non-overlap while preserving sufficient training diversity. (2) Since all splits were manually verified, adding more seeds would require manually checking more instances, which is time-consuming and error-prone. (3) Evaluation cost is substantial: one seed per template already requires ~2.5 hours on 4 parallel Android emulators (24 CPU cores + 2×H100-80G). Using three seeds significantly increases this cost, and enlarging the test set further would make RL evaluation prohibitively expensive during development. For comparison, the original AndroidWorld leaderboard uses only a single seed; our benchmark already evaluates on three to improve robustness.
>
> We acknowledge that the current scale is not ideal, and expanding to more templates and apps is an important direction. However, scaling the benchmarks reliably remains an open community challenge requiring better task generators and automated verification that extend beyond the scope of this work. Instead, our goal is to establish the first RL generalization benchmark for mobile agents.
>
>
>
>
> > W3. Test data should not be used for analyzing training dynamics, e.g., in Fig 3, 4. This serves as a form of information leakage in the experiment, where we are no longer using the test data to evaluate true generalization of the approach to held-out, IID data.
>
> We appreciate the reviewer’s concern; however, there may be a misunderstanding about how the test set is used in our experiments. We fully agree that test data must not influence training or model selection. In our setup, the agent is trained solely on the training set, and the test curves in Fig. 3–4 were computed only after training had fully completed. The plots are generated by evaluating saved checkpoints after all training, rather than querying the test set during training. These curves are included only to **illustrate the divergence between training performance and generalization performance**; they play no role in hyperparameter choices, early stopping, or any training decision. Consequently, the test set remains a strictly held-out evaluation set, and no information leakage occurs.
>
> This usage is consistent with established practice in RL generalization research. Influential benchmarks[1,2,3]  also plot both train and test curves over training steps. In those works, the test set is held-out like ours, and plotting both curves is used to diagnose overfitting and characterize **generalization gaps**, not to guide training. Our figures serve the same diagnostic purpose and do not compromise the validity of the test set as a held-out evaluation.
>
> To avoid future confusion, in the revised version we explicitly state that the test curves are computed post-training between Line 426
>
> [1] Quantifying Generalization in Reinforcement Learning. OpenAI. ICML 2019
> [2] Why Generalization in RL is Difficult: Epistemic POMDPs and Implicit Partial Observability. UC Berkeley. NeurIPS 2021
> [3] The Generalization Gap in Offline Reinforcement Learning. FAIR. ICLR 2024
>
> > W1. The intro has a ton of citations in it, these should really only be in the related work unless they are central to the core argument of the paper
>
> Thank you for the suggestion. We have revised the introduction by removing non-essential citations and retaining only those directly supporting the core motivation of our work.

---

> ### Author Response · Authors · 2025-11-26
>
> > Q2. How is task difficulty determined? This is coming from AndroidWorld, but how do they define it?
>
> Yes. The task difficulty is defined by the original AndroidWorld paper. As described therein, the authors recruited six volunteers with proficient programming skills to assess each task’s difficulty, expected duration, and category. Based on this assessment, they assigned a difficulty level and estimated the number of steps required to complete each task, using the available action space. However, the paper does not provide an explicit, formal rule for how difficulty labels were annotated. From our analysis of the predefined task templates, difficulty appears strongly correlated with factors such as the estimated number of steps required and the number of apps involved in completing the task.

---

### Author Response · Authors · 2025-11-26
**To All**

We thank reviewers for their positive feedback. Our paper makes the following contributions:

* **First study of generalization in RL for mobile agents**: We (1) formalize mobile-agent generalization as a constrained MDP problem and (2) introduce AndroidWorld-Generalization, a new benchmark with three well-defined unseen regimes.
* **First open-source RL training system for mobile agents**: We develop a complete training and rollout-collection infrastructure integrating GRPO with significant system-level optimizations tailored to mobile environments.
* **First empirical analysis of generalization challenges**: Through extensive experiments, we identify the difficulties of generalization and demonstrate few-shot test-time adaptation as a promising and practical direction.

This work intentionally focuses on problem formulation, benchmark design, open-source system development, and empirical insights, **rather than proposing new algorithms**. These foundational components aim to motivate and support future algorithmic advances in this emerging research area.


All reviewers recognized the significance of our contributions, particularly the importance of formally defining the RL generalization problem for mobile agents and demonstrating the practical effectiveness of RL in this domain. They highlighted the novelty and clarity of our benchmark design [yrmE], the usefulness of our open-source system infrastructure [UZoD, yrmE], and the overall clarity of writing [UZoD].

The reviewers’ main concerns focused on four areas: (1) the need for additional detail on the task-instance generation process in the benchmark, (2) clarification of the term “training framework,” (3) further justification of certain experimental design choices, and (4) the inclusion of PPO results to validate the generality of our RL training system.


In response, the rebuttal and revised paper incorporate several major updates. We added PPO results as requested [UZoD, yrmE], moved key benchmark-design details from Appendix Section C into Section 3.2, and reformatted Table 1 to more clearly present the three unseen regimes [pnBC]. To avoid conceptual ambiguity, we replaced “training framework” with “RL training system,” emphasizing our contribution on system engineering rather than algorithmic innovation. We also added a paragraph in the Introduction discussing system-level limitations in the current landscape and improved Table 2 to more clearly show performance gains over prior reproducible baselines [UZoD]. Additional explanations were provided to justify experimental choices and clarify GRPO-related design decisions [yrmE]. All revisions are highlighted in blue and will be removed in the camera-ready version.

---

### Author Response · Authors · 2025-12-02
**Summary of Rebuttal for the New Area Chair**

We have made the following revisions to address the reviewers’ concerns. All changes are highlighted in blue in the revised paper:

1. **Clarifying task-instance generation (Reviewer pnBC)**:
We moved the technical details and concrete examples of train-task instance generation from Appendix Section C to Section 3.2 of the main paper and reformatted Table 1 for clarity.

2. **Clarifying contributions and providing context (Reviewer UZoD)**:
Because the reviewer appeared unfamiliar with the emerging field of LLM-based mobile agents, we added brief background to highlight the missing foundations—benchmarking, reproducibility, and open-source infrastructure. We clarified that our contribution lies in building these foundations rather than proposing a new algorithm, and we strengthened the comparison with the original AndroidWorld to emphasize novelty.

3. **Correcting terminology to avoid ambiguity (Reviewer UZoD)**:
We replaced “end-to-end RL framework” with “RL training system” to remove ambiguity that led to the reviewer’s incorrect interpretation of our contribution.

4. **Demonstrating algorithm-agnostic design (Reviewers UZoD and yrmE)**:
Since some reviewers interpreted our system as tied to a specific GRPO algorithm, we added PPO results to show that the RL training system is algorithm-agnostic.

5. **Improving experimental clarity (All reviewers)**:
Although the experiment setup was already described, we expanded several details to prevent further misunderstanding and to make the design choices fully transparent.

We note that the reviewers have not posted post-rebuttal follow-ups, likely due to the system rollback occurring shortly after our rebuttal submission. However, our rebuttal addresses **all raised concerns**. We appreciate your time and consideration.

---

### Meta-Review · Area_Chair_7c4S · 2025-12-05

**Summary:**

The reviewers generally recognized the value of the problem formulation and the open-source contribution but raised several concerns regarding clarity, novelty, and experimental design.

Reviewer pnBC found the formatting of Table 1 confusing regarding the training/testing splits, questioned the statistical significance of the small test sizes, and flagged potential data leakage where test data appeared to be used in training dynamics plots.

Reviewer UZOD worries about the novelty of the work compared to the original AndroidWorld, argued that the proposed "RL training framework" was too limited by only implementing GRPO, and sought clarification on how the baselines were trained to ensure fair comparison.

Reviewer yrmE pointed out the limited scale of the benchmark (20 apps) and the underdeveloped nature of the few-shot adaptation experiments, while also asking about the justification for choosing GRPO over other algorithms like PPO.

**Reviewer Concerns:**

The authors provided a rebuttal that appears to address some of the concerns, though the limitation on scale remains.

 Regarding Reviewer pnBC's confusion, the authors moved generation details to the main text and reformatted Table 1 to clarify the non-overlapping splits, while also explaining that test curves were generated via post-training evaluation to refute leakage claims.

For Reviewer UZOD's concerns on novelty and scope, the authors clarified the distinction between their RL-ready system and the evaluation-only AndroidWorld, renamed "framework" to "system" to reduce ambiguity, and added PPO results to demonstrate the system is algorithm-agnostic.

 The authors also addressed Reviewer yrmE's questions by adding PPO comparisons and justifying the limited scale as an inherent environment constraint rather than a methodological flaw.

The concern regarding the small test set size raised by pnBC and yrmE remains outstanding.  The authors provided a justification based on the high cost of manual verification and computational resources required for reliable non-overlapping splits.

**Reviewer Scores:**

Reviewer pnBC (Score 4): Their primary confusion regarding data splits and leakage was effectively addressed by the authors' clarifications and the restructuring of the manuscript. However, the concern regarding the small size of the test set remains. Since the authors attributed this limitation to experimental costs, a justification that may not fully satisfy the reviewer, it is likely they would be inclined to maintain their current score.

Reviewer UZOD (Score 2): This reviewer maintained that the benchmark's reliance on the existing AndroidWorld environment limits its novelty. Although the authors successfully addressed other methodological concerns, this fundamental objection regarding novelty persists. Consequently, while their score might improve slightly (e.g., to a 4), they may still lean towards rejection.

Reviewer yrmE (Score 6): This reviewer was already positive. The inclusion of additional PPO experiments and the justifications regarding the benchmark's scale would likely solidify their support or slightly increase their confidence in recommending acceptance.

---

### Decision · Program_Chairs · 2026-01-26

Reject